# Decoupling Policy Improvement and Conservatism in Offline Reinforcement Learning

## Abstract

Offline reinforcement learning (RL) methods suffer from *extrapolation error*. However, current solutions face a dilemma - static constraints are over-conservative, while naive dynamic references perilously couple policy improvement with conservatism, which can lead the reference itself to an out-of-distribution (OOD) target. Our core insight is that these two objectives must be decoupled. In this paper, we introduce the **C**overage-**A**ware **R**eference Policy (CAR), which instantiates this principle via a propose-and-verify mechanism: the learned policy proposes actions, and a verifier confirms data support before augmenting a reference policy. This tractably creates a progressively improving reference with theoretical coverage guarantees. Finally, CAR establishes state-of-the-art (SOTA) results on offline RL benchmarks and strong results in online fine-tuning.

## 1 Introduction

Offline Reinforcement Learning (RL) (Lange et al., 2012) aims to learn effective policies from a fixed dataset. Its core challenge lies in the distributional shift between the learned policy and the data-collection policy, which leads to severe extrapolation error: an overestimation of the value for out-of-distribution (OOD) actions, thereby causing standard algorithms to fail (Fujimoto et al., 2019). To solve this problem, mainstream methods constrain the learned policy by imposing conservatism, which can be divided into two categories: directly applying behavior regularization to keep it close to the behavior policy (Fujimoto et al., 2019; Fujimoto & Gu, 2021; Jiang et al., 2023), or penalizing the Q-values of OOD actions (Kumar et al., 2020; Yang et al., 2022). However, although behavior regularization is an essential method of mitigating the problem, strictly regularizing the policy to be near the static behavior policy often leads to over-conservatism (Wu et al., 2022). This greatly limits the policy's performance ceiling, especially when the dataset is generated by a suboptimal policy (Kumar et al., 2019).

To move beyond a static behavior policy, more advanced approaches constrain learning towards an improving reference policy. A common strategy is to use the learned policy itself as the reference, as seen in ABM (Siegel et al., 2020), iTRPO (Zhang & Tan, 2024), and CPI (Ma et al., 2024). This, however, creates a fundamental dilemma by bundling the agent of improvement with the enforcer of conservatism. The learned policy is expected to simultaneously drive performance upward while also serving as a in-support anchor.

This inherent conflict of roles is untenable. The assumption that the learned policy will remain in-support throughout training—a practical form of the theoretically restrictive all-policy concentrability assumption (Xie et al., 2021)—is frequently violated. Even minor parameter updates can cause significant distributional shifts (Zhang & Tan, 2024). When this occurs, the reference policy becomes an OOD learning target, which nullifies the conservative principle and steers the agent towards extrapolation error. Attempts to mitigate this by adding a second constraint (e.g., behavior regularization) merely transform the problem into a complex, and sometimes infeasible, multi-objective optimization (Achiam et al., 2017). The root of this issue is the forced choice between uniform pessimism (imitating the suboptimal dataset) and misguided optimism (tracking an OOD target). This dilemma is a direct consequence of refusing to decouple the intertwined goals of policy improvement and conservatism.

Our key insight is that a better trade-off between policy improvement and conservatism can be achieved by decoupling these two objectives. To this end, we propose the **C**overage-**A**ware **R**eference

Policy (CAR), a framework that operationalizes this decoupling through a novel propose-and-verify mechanism. The *improvement* is driven by the learned policy, which proposes high-performing actions. The *conservatism* is enforced by a verification step that guarantees these actions are within the data's support before they are used to augment a reference policy. This process tractably creates a progressively improving reference policy that avoids the limitations of a static dataset. Theoretically, we prove that CAR maintains a formal coverage guarantee. Empirically, CAR outperforms state-of-the-art (SOTA) methods by a large margin with only a slight increase in computational overhead.

This work has three main contributions:

- We propose progressively improving behavior regularization while ensuring in-distribution in offline RL. This is achieved through a novel propose-and-verify sampling mechanism to augments the static behavior policy.

- We theoretically show that CAR comes with a coverage guarantee, which aids learning by providing a performance lower bound.

- Compared to several strong baselines, CAR achieves SOTA results on standard offline RL benchmarks and demonstrates strong results in online fine-tuning tasks.

## 2 RELATED WORK

Our work belongs to the family of policy constraint methods in offline RL. These methods constrain the learned policy to remain close to a specific reference policy by penalizing their divergence, which is formulated as a regularization term in the policy optimization objective.

**Behavior policy constraint.** Constraining the policy towards the behavior policy is an effective method for mitigating extrapolation error. BCQ (Fujimoto et al., 2019) learns a parameterized generative model of the behavior policy and trained to perturb randomly generated actions. Many methods employ a divergence penalty for regularization. For instance, TD3+BC (Fujimoto & Gu, 2021) directly clones behavior using Mean Squared Error. BRAC (Wu et al., 2019) utilizes Kullback-Leibler (KL) Divergence and BEAR (Kumar et al., 2019) employs Maximum Mean Discrepancy. SPOT (Wu et al., 2022) and CPED (Zhang et al., 2023) are also based on KL divergence, but they achieve better performance by using generative models to get a better probability estimate of the behavior policy.

**Reference policy constraint.** These methods leverage a improved reference policy for less conservative. ABM (Siegel et al., 2020) treat the learned policy as a successful prior, which acts as a constraint and completely abandoning behavior policy, but this approach losses conservatism. Addressing this drawback, STR (Mao et al., 2023) builds upon ABM by incorporating importance sampling to avoid guidance from OOD actions of the reference policy. iTRPO (Zhang & Tan, 2024) employs the trust-region concept, uses both behavior policy and learned policy to construct two constraints. Following a similar concept with iTRPO, CPI (Ma et al., 2024) uses the learned policy to guide performance improvement, while retaining a constraint against the behavior policy to ensure conservatism. Existing methods reveals a fundamental dilemma that are inherently either uniformly pessimistic or optimistic. In this work, we address this in a flexible way.

## 3 PRELIMINARY

In reinforcement learning, we generally modeled the environment as a Markov Decision Process (MDP) $\mathcal{M} = (\mathcal{S}, \mathcal{A}, P, R, \gamma, \rho_0)$, with state space $\mathcal{S}$, action space $\mathcal{A}$, transition dynamics $P : \mathcal{S} \times \mathcal{A} \to \Delta(\mathcal{S})$, reward function $R : \mathcal{S} \times \mathcal{A} \to [0, R_{\max}]$, discount factor $\gamma \in [0, 1)$, and initial state distribution $\rho_0$, where $\Delta(\cdot)$ denotes the probability simplex. The agent interacts with the environment and seeks a policy $\pi : \mathcal{S} \to \Delta(\mathcal{A})$ to maximize the expected discounted return $J(\pi) := \mathbb{E}[\sum_{t=0}^{\infty} \gamma^t r_t | \pi \in \Pi]$ for all $t \geq 0$. The objective of reinforcement learning is to find an optimal policy: $\pi^* := \arg\max_\pi J(\pi)$. It will be useful to define the *policy-specific* state-action value function (Q-value function, $Q^\pi : \mathcal{S} \times \mathcal{A} \to \mathbb{R}$) as $Q^\pi(s, a) = \mathbb{E}[\sum_{t=0}^{\infty} \gamma^t r_t | s_0 = s, a_0 = a, a_{1:\infty} \sim \pi]$, which represents the expected discounted return starting from state $s$ and taking action $a$, and

subsequently following policy $\pi$. $Q^\pi(s, a)$ is the unique fixed point of the Bellman expectation operator $\mathcal{T}^\pi : \mathbb{R}^{\mathcal{S} \times \mathcal{A}} \to \mathbb{R}^{\mathcal{S} \times \mathcal{A}}$, defined via bootstrapping as:

$$(\mathcal{T}^\pi Q^\pi)(s, a) = R(s, a) + \gamma \mathbb{E}_{s' \sim P(\cdot|s,a)}[Q^\pi(s', \pi)], \tag{1}$$

where $Q^\pi(s', \pi)$ is shorthand for $\mathbb{E}_{a' \sim \pi(\cdot|s')}[Q^\pi(s', a')]$.

Offline RL addresses the problem of learning an effective policy from a previously collected dataset, denoted as $\mathcal{D} = \{s_i, a_i, r_i, s'_i\}_{i=1}^n$. In large and complex state-action spaces, function approximation is typically employed to model $Q^\pi$. For example, an iterative approach commonly initializes $Q_0$ arbitrarily, and iteratively updates the parameterized policy at update step $t$ by maximizing the Q-value to maximize the estimated return: $\pi_{t+1} = \arg\max_\pi \mathbb{E}_{s \sim \mathcal{D}} \left[ \mathbb{E}_{a \sim \pi(\cdot|s)}[Q_t(s, a)] \right]$, Alternatingly, the parameterized function can be updated:

$$Q_{t+1} = \arg\min_Q \mathbb{E}_{(s,a,r,s') \sim \mathcal{D}} \left[ Q(s, a) - r - \gamma Q_t(s', \pi_{t+1}) \right]^2. \tag{2}$$

Unlike online RL, the learning process in the offline setting is strictly limited to this static dataset. This fundamental limitation brings challenges, particularly the *distribution shift* between the learned policy $\pi_t$ and the behavior policy $\pi_\beta$. To mitigate the extrapolation error, a solution is to constrain the learned policy imitate the behavior policy using divergence measure, such as Kullback-Leibler (KL) Divergence (Wu et al., 2019). However, imitating the suboptimal $\pi_\beta$ inevitably results in limited performance. More advanced approaches change the regularization towards an current learned policy, which also named as a *reference policy* (Ma et al., 2024) or a *prior* (Siegel et al., 2020). However, these methods ignore that the learned policy itself can produce OOD actions, defeating the very purpose of policy constraint by making the learning objective unreliable.

## 4 THE METHOD

This section provides a detailed introduction to **C**overage-**A**ware **R**eference Policy(CAR). Concretely, CAR involves a new reference policy for policy constraint, which leverages information from the learned policy for improvement while ensuring it remains within the dataset's support. To realize this new reference policy tractably, we employ a mechanism to get sample from CAR. This mechanism uses a propose-and-verify sampling process to sample from this reference policy, and then reshapes the dataset's action distribution by attaching these samples. Finally, we use explicit density regularization methods to arrive at the practical algorithm.

### 4.1 COVERAGE-AWARE REFERENCE POLICY

Our goal is to design a reference policy that incorporates information from the learned policy $\pi_t$ (which reflects the current improving signal) and the behavior policy $\pi_\beta$ (which provides support information). The learned policy $\pi_{t+1}$ is then constraint to be close to this reference policy. The policy learning objective can be formulated as:

$$\max_\pi \mathbb{E}_{s \sim \mathcal{D}} \left[ \mathbb{E}_{a \sim \pi(\cdot|s)}[Q_t(s, a)] - \lambda D_{KL}(\pi \| \pi_{ref}) \right], \tag{3}$$

where $\pi_{ref}$ is usually chosen to be a combination of the learned policy $\pi_t$ and the behavior policy $\pi_\beta$: $\pi_{mix}(a|s) = \tau \pi_t(a|s) + (1 - \tau)\pi_\beta(a|s)$, $\tau$ is the mixing coefficient. Note that the reference policies employed by prior works, such as CPI (Ma et al., 2024), can be considered a special case of this mixed policy.

While this new constraint between the target policy $\pi$ and the reference policy $\pi_{ref}$ simplifies the optimization problem, it still suffers from OOD actions from $\pi_t$ (i.e., $\exists a, \pi_{ref}(a|s) > 0$ s.t. $\pi_\beta(a|s) \to 0$). We empirically verify this phenomenon in Section 5.3. Therefore, our next step is to refine the mixed policy to guarantee it stays within the dataset's support.

To this end, we define the coverage-aware mixed policy $\pi_{mix}$ based on the support information of the behavior policy. At iteration $t$, the mixed policy is defined as:

$$\pi_{mix}(a|s) = \tau \hat{\pi}_t(a|s) + (1 - \tau)\pi_\beta(a|s), \text{ where } \hat{\pi}_t(a|s) = \frac{1}{Z_t(s)} \pi_t(a|s) \mathbb{I}(\pi_\beta(a|s) \geq \epsilon), \tag{4}$$

$Z_t(s) = \sum_a \pi_t(a|s) \mathbb{I}(\pi_\beta(a|s) \geq \epsilon)$ serves as a normalization term, $\mathbb{I}(\cdot)$ is an indicator function that outputs 1 if the condition is met and 0 otherwise, and $\epsilon \in [0, 1]$ is a constant support threshold. Note that we adopt the convention that $0/0 = 0$, and a non-zero value divided by 0 is infinity.

Formally, $\pi_{mix}(a|s)$ achieves coverage-awareness by using the support of the behavior policy (where $\pi_\beta(a|s) > \epsilon$) to mask the learned policy $\pi_t$. This process yields a purified policy, $\hat{\pi}_t(a|s)$, that preserves the improvements of $\pi_t$ while eliminating risky OOD actions. The final $\pi_{mix}(a|s)$ is then a weighted combination of this purified policy and $\pi_\beta$.

However, estimating this mixed policy is intractable because the normalization term $Z_t(s)$ requires a summation over the infinite action space. To overcome this challenge, in the next section, we design a novel mechanism to avoid direct estimation of this probability distribution.

## 4.2 SAMPLING FROM CAR

Directly estimating the intractable distribution $\pi_{mix}(a|s)$ is infeasible. Our core insight is that sampling from this distribution can be made tractable by strategically defining the mixing coefficient, $\tau_t(s)$. Specifically, we set $\tau_t(s)$ to be the probability that an action drawn from the learned policy $\pi_t$ is in-support:

$$\tau_t(s) \triangleq \mathbb{E}_{a \sim \pi(a|s)}[\mathbb{I}(\pi_\beta(a|s) \geq \epsilon)] \equiv P_{accept,t}(s) \tag{5}$$

This definition gives the mixing coefficient a clear probabilistic interpretation and directly motivates a propose-and-verify sampling mechanism. The procedure first *proposes* an action by sampling from $\pi_t$. It then *verifies* its support. If the action is accepted (an event with probability $P_{accept,t}(s)$), this step corresponds to sampling from the masked learned policy component, $P_{accept,t}(s)\hat{\pi}_t(a|s)$. Conversely, if the action is rejected, a new action is drawn from the behavior policy $\pi_\beta$, corresponding to sampling from the second component of the mixture, $(1 - P_{accept,t}(s))\pi_\beta(a|s)$. This elegant design allows us to sample from the desired hybrid distribution without ever needing to compute it explicitly.

This design yields two significant advantages. First, it is computationally tractable, as it circumvents the need to explicitly compute the normalization constant $Z_t(s)$. Second, it transfers the locus of control over conservatism from an abstract, manually-tuned hyperparameter, $\tau$, to the more interpretable and semantically meaningful support threshold, $\epsilon$. This allows for more intuitive safety tuning: a stricter requirement (i.e., a higher $\epsilon$) naturally reduces the acceptance probability of proposed actions, thereby causing the policy to fall back on the conservative behavior policy, $\pi_\beta$. Ultimately, this mechanism efficiently generates a stream of action samples that are simultaneously in-support and indicative of policy improvement. These samples then serve as training data to learn an explicit density model of the mixed policy via standard supervised learning.

While the mixed policy remains within the dataset's support, it still can be overly conservative. Specifically, when $\pi_t$'s proposed actions are rejected, causing $\pi_{mix}$ to fall back excessively towards $\pi_\beta$, this can lead to potentially unstable training or slow convergence. A more effective design should fully utilize and retain all acquired knowledge of learning process. To address this limitation, we further modify the mixed policy at the current iteration $t$ as:

$$\pi_{mix,t}(a|s) = \tau_t(s)\hat{\pi}_t(a|s) + (1 - \tau_t(s))\pi_{mix,t-1}(a|s), \tag{6}$$

where $\hat{\pi}_t(a|s) = \dfrac{1}{Z_t(s)}\pi_t(a|s)\mathbb{I}(\pi_\beta(a|s) \geq \epsilon)$, $\tau_t(s) \triangleq Z_t(s) = \sum_a \pi_t(a|s)\mathbb{I}(\pi_\beta(a|s) \geq \epsilon)$. (7)

Specifically, we define $\pi_{mix,0}(a|s) = \pi_\beta(a|s)$. This improved design incorporates the mixed policy with historical mixed policy $\pi_{mix,t-1}(a|s)$. When the proposal of $\pi_t(a|s)$ is rejected, it will fall back to the policy from the previous timestep, rather than a much earlier behavioral policy. In the next section, we can prove that $\pi_{mix,t}(a|s)$ is always in-support. Intuitively, we should use this mixture policy, which avoids acting OOD actions, for both evaluation and regularization in Eq.(3) and Eq.(2).

## 4.3 ANALYSIS

This section is dedicated to the theoretical analysis of the key properties of our proposed mixed policy. First, our analysis demonstrates that the mixed policy is guaranteed to be in-support, with its

probability distribution always covered by the support of the behavior policy. Second, we provide a performance lower bound for the mixed policy that incorporates general function approximation error. Our analysis shows that this performance lower bound is theoretically guaranteed and is directly controlled by the support threshold $\epsilon$.

We begin by providing a coverage guarantee to prove that the mixed policy always remains within the support of the behavior policy.

**Theorem 1.** *(Coverage guarantee). For any $\pi_{mix,t}$ in Eq.(6) and let $\epsilon > 0$ be the support threshold, the coverage condition is bounded above for any $t \geq 1$:*

$$\max_{s,a} \frac{\pi_{mix,t}(a|s)}{\pi_\beta(a|s)} \leq \frac{1}{\epsilon} \tag{8}$$

A detailed proof can be found in Appendix A. Theorem 1 proves that for any state $s$ and action $a$, the probability ratio between mixed policy $\pi_{mix,t}$ and the behavior policy $\pi_\beta$ has a constant upper bound. This conclusion provides a formal guarantee that mixed policy remains within the support of the behavior policy, even when the learned policy $\pi_t$ is generating OOD actions.

A key departure from many offline RL methods is their reliance on stringent assumptions regarding data coverage, such as requiring the dataset to provide sufficient support for a wide range of policies (Mao et al., 2023; Zhang & Tan, 2024). Our method, in contrast, imposes a constructive constraint directly on the policy itself. This design ensures that the resulting mixed policy is provably constrained to the dataset's support, irrespective of the dataset's intrinsic quality. This explicit support guarantee is the cornerstone of our theoretical analysis, enabling the derivation of a meaningful performance lower bound for the mixed policy, which we present in Theorem 2.

**Theorem 2.** *Under Theorem 1 and approximate completeness assumption, for any $\pi_{mix,t}$ in Eq.(6) and any $\epsilon > 0$, apply Bellman Residual (Error) Minimization (Xie et al., 2021) and then with probability at least $1 - \delta$ the following bound holds:*

$$J(\pi_\beta) - J(\pi_{mix,t}) \leq \frac{V_{\max}\sqrt{-\log\epsilon}}{\sqrt{2}(1-\gamma)} + (1 + \frac{|\mathcal{S}|}{\epsilon})\frac{\sqrt{\epsilon_b}}{1-\gamma}, \tag{9}$$

*where $\sqrt{\epsilon_b} \leq \mathcal{O}\left(V_{\max}\sqrt{\frac{\log |\mathcal{F}||\Pi|/\delta}{n}}\right)$ is mean-squared bellman error.*

An analysis of the performance lower bound in Theorem 2 reveals that the error terms are principally controlled by the support threshold $\epsilon$. The bound comprises two primary error sources: the first term quantifies the performance deviation between the mixed policy and the behavior policy, while the second captures the inherent error from function approximation. The choice of $\epsilon$ introduces a clear trade-off. Increasing $\epsilon$ towards its upper limit of 1 mitigates or even nullifies the first error term by forcing the mixed policy to closely mimic the behavior policy. While the function approximation error is also diminished with a higher $\epsilon$, it cannot be entirely eliminated. A detailed proof of Theorem 2 is provided in Appendix A.

Prior methods, e.g., Ma et al. (2024), can be interpreted as directly using the learned policy to augment the behavior policy without a verification mechanism. This approach is analogous to the limiting case of our framework where $\epsilon \to 0$. In this limit, key error terms in the theoretical performance lower bound diverge, rendering the guarantee vacuous. CAR, by contrast, adaptively modulates the influence of the learned policy via its propose-and-verify mechanism. This ensures that the error terms remain bounded, thereby preserving a non-vacuous and meaningful performance guarantee.

### 4.4 PRACTICAL IMPLEMENTATION

The practical CAR algorithm is designed to be as simple as possible. Our method is built upon SPOT (Wu et al., 2022) with minor modifications, retaining algorithmic simplicity and improving computational efficiency.

**Explicit Density Estimator.** We explicitly learn a density estimator to model the dataset's policy. Prior works have utilized generative models, leveraging their ability to capture complex distributions. For instance, SPOT uses a Variational Autoencoder (VAE) (Wu et al., 2022), CPED uses a Flow-GAN (Zhang et al., 2023), and OSC employs a diffusion model (Gao et al., 2025). While more

---

**Algorithm 1** The pseudocode of CAR

---

**Input**: The offline dataset $\mathcal{D} = \{(s, a, r, s')\}$

1: Initialize VAE $\hat{\pi}_\beta$ with parameters $\psi$ and $\varphi$, policy network $\pi_\phi$, critic network $Q_\theta$
    **// Warm-up Training**
2: **for** $t = 1$ to $T_1$ **do**
3:    Sample mini-batch of transitions $(s, a, r, s') \sim \mathcal{D}$
4:    Update $\psi, \varphi$ by maximizing Eq.(10), Update $\theta$ by Eq.(12), Update $\phi$ using $\hat{\pi}_\beta$ by Eq.(11)
5: **end for**
6: Initialize VAE $\hat{\pi}_{mix}$ with parameters $\psi' \leftarrow \psi$ and $\varphi' \leftarrow \varphi$ and empty dataset $\mathcal{D}' = \emptyset$
    **// RL Training**
7: **for** $t = T_1$ to $T_2$ **do**
8:    Sample mini-batch of transitions $(s, a, r, s') \sim \mathcal{D}$
9:    Update $\theta$ by Eq.(12)
10:    Sample action $\hat{a} = \pi_\phi(s)$ to update $\phi$ by Eq.(11)
11:    Calculate $-\log \hat{\pi}_\beta(\hat{a}|s)$
12:    Add $(s, \hat{a})$ to $\mathcal{D}'$ if $-\log \hat{\pi}_\beta(\hat{a}|s) \leq \epsilon_{ood}$
13:    Sample mini-batch state-action pairs $(s, a) \sim \mathcal{D} \cup \mathcal{D}'$ to update $\psi', \varphi'$ by Eq.(10)
14: **end for**

---

advanced generative models could model the behavior policy more accurately, we still choose the more widely-used VAE to ensure a fair comparison with baselines. A conditional VAE parameterized by $(\psi, \varphi)$ is trained to model the mixed policy by maximizing the Evidence Lower Bound (ELBO) with respect to the data samples $(s, a) \in \mathcal{D} \cup \mathcal{D}'$. The ELBO loss for a single state-action pair is defined as:

$\log \pi(a|s) \geq \mathbb{E}_{q_\varphi(z|a,s)} [\log p_\psi(a|z, s)] - D_{KL}(q_\varphi(z|a, s) \| p(z|s)) \stackrel{\text{def}}{=} -\mathcal{L}_{\text{ELBO}}(s, a; \psi, \varphi)$, where $p(z|s)$ is a prior set as a standard Gaussian. Minimizing ELBO loss can update the density estimator:

$$(\psi_t, \varphi_t) = \arg\min_{\psi, \varphi} \mathbb{E}_{(s,a)\sim\mathcal{D}\cup\mathcal{D}'} \mathcal{L}_{\text{ELBO}}(s, a; \psi, \varphi). \tag{10}$$

Then ELBO loss approximates the negative log-probability: $-\log \hat{\pi}_{mix,t}(a|s) \approx \mathcal{L}_{\text{ELBO}}(s, a; \psi_t, \varphi_t)$.

**Coverage-Aware Reference Policy.** A key aspect of implementing the mixed reference policy is the condition within the indicator function. We use a VAE to model $\hat{\pi}_\beta$ and verify this condition for a given data: $-\log \hat{\pi}_\beta(\hat{a}|s) \leq \epsilon_{ood}$, where $\epsilon_{ood}$ is a hyperparameter. If the condition is met, we add this data to an auxiliary dataset $\mathcal{D}'$, which serves as a buffer for $\pi_{mix,t}$. For computational efficiency, we only store a single batch of data at a time rather than processing the entire offline dataset $\mathcal{D}$.

**Overall algorithm.** Our algorithm is based on SPOT, but unlike it, we don't pre-train the density estimator. Instead, we model the behavior policy using Eq.(10) during the warm-up phase. RL algorithm follow most prior methods (Tarasov et al., 2023), a deterministic policy network is trained to maximize the Q-value function while constrained. Thus, we have the following policy objective, with parameters $\phi$, at iteration $t$:

$$\phi_{t+1} = \arg\max_\phi \mathbb{E}_{s\sim\mathcal{D}} [Q_{\theta_t}(s, \pi_\phi(s)) + \lambda \log \hat{\pi}_{mix,t}(\pi_\phi(s)|s)] . \tag{11}$$

where $\lambda > 0$ is a regularization coefficient. During the warm-up phase, the second term is simply $\log \hat{\pi}_\beta(\cdot|s)$. During the RL training phase, we use the learned policy $\pi_{\phi_{t+1}}$ to generate data for updating the mixed policy model $\hat{\pi}_{mix,t+1}$. The Q-value function parameterized by $\phi$ is updated by minimizing the temporal difference (TD) error:

$$\theta_{t+1} = \arg\min_\theta \mathbb{E}_{(s,a,r,s')\sim\mathcal{D}} \left[ Q(s, a) - r - \gamma Q_{\bar{\theta}_t}(s', \hat{\pi}_{\bar{\phi}_{t+1}}(s')) \right]^2 . \tag{12}$$

where $\bar{\theta}$ and $\bar{\phi}$ both parameters of the target network. Although theoretically we should use the mixed policy for policy evaluation, the final experiment proved that using the learned policy would achieve better performance in Section 5.4.

Integrating all these components, our full algorithm is summarized in Algorithm 1. For simplicity, standard components like the target network and delayed actor updates are omitted.

Table 1: The mean and standard deviation of averaged normalized scores after 1M steps (3M for RORL) of offline RL training, averaged over 5 seeds and the final 10 evaluations (final 4 for Antmaze). We bold the highest scores in each task.

| | | CQL | TD3+BC | reBRAC | SPOT | RORL | CPI | iTRPO | CAR(ours) |
|---|---|---|---|---|---|---|---|---|---|
| halfcheetah | r | 31.3±3.5 | 11.8±0.5 | 14.5±2.5 | 24.7±0.3 | 28.5±0.8 | 29.7±1.1 | 27.4±0.3 | **39.2±0.6** |
| | m | 46.9±0.4 | 48.2±0.1 | 64.0±0.7 | 61.3±0.4 | 66.8±0.7 | 64.4±1.3 | 56.2±1.1 | **66.9±1.7** |
| | m-r | 45.3±0.3 | 44.6±0.1 | 51.2±0.3 | 50.0±2.5 | **61.9±1.5** | 54.6±1.3 | 55.0±0.5 | 60.5±2.6 |
| | m-e | 95.0±1.4 | 91.7±1.9 | 103.8±3.0 | 93.1±2.2 | **107.8±1.1** | 94.7±1.1 | 94.4±0.3 | 97.2±0.9 |
| | e | 97.3±1.1 | 96.7±0.5 | 100.1±1.3 | 93.9±0.2 | **105.2±0.7** | 96.5±0.2 | 95.2±0.1 | 99.2±1.3 |
| hopper | r | 5.3±0.6 | 9.1±1.6 | 10.6±3.5 | 26.0±9.9 | 31.4±0.1 | 29.5±3.7 | 31.6±0.1 | **32.7±0.1** |
| | m | 61.9±6.4 | 59.0±0.7 | 102.3±0.2 | 93.1±7.0 | **104.8±0.1** | 98.5±3.0 | 98.5±0.6 | 103.7±0.1 |
| | m-r | 86.3±7.3 | 65.4±14.0 | 95.0±6.5 | 99.1±1.7 | 102.8±0.5 | 101.7±1.6 | 101.2±0.7 | **103.9±0.2** |
| | m-e | 96.9±15.1 | 101.6±2.0 | 109.5±2.3 | 112.0±0.9 | **112.7±0.2** | 106.4±4.3 | 110.8±0.3 | 110.2±4.5 |
| | e | 106.5±9.1 | 110.2±1.9 | 110.2±1.2 | 111.4±0.4 | **112.8±0.2** | 112.2±0.5 | 111.3±0.1 | 111.4±1.6 |
| walker2d | r | 5.4±1.7 | 1.4±1.0 | 1.7±3.1 | 5.8±0.2 | 21.4±0.2 | 5.9±1.7 | 5.7±4.7 | **21.7±0.1** |
| | m | 79.5±3.2 | 83.8±0.3 | 85.8±0.8 | 82.4±2.1 | **102.4±1.4** | 85.8±0.8 | 85.0±0.4 | 94.7±0.7 |
| | m-r | 76.8±10.0 | 80.3±1.8 | 84.3±2.3 | 91.7±1.0 | 90.4±0.5 | 91.8±2.9 | 94.2±3.1 | **104.0±0.7** |
| | m-e | 109.1±0.2 | 110.1±0.3 | 111.9±0.4 | 110.3±0.1 | **121.2±1.5** | 110.9±0.4 | 110.6±0.4 | 115.3±0.5 |
| | e | 109.3±0.1 | 110.2±0.1 | 109.9±0.2 | 109.9±0.2 | 115.4±0.5 | 110.6±0.1 | 111.8±1.9 | **117.8±0.2** |
| MuJoCo | | 1052.8±60.4 | 1024.1±26.8 | 1154.8±28.3 | 1164.6±29.2 | **1285.5±10** | 1193.2±24 | 1188.9±14.6 | 1278.2±15.6 |
| antmaze | u | 92.8±1.9 | 70.8±39.2 | 97.8±1.5 | 95.6±2.7 | 96.7±1.9 | 98.8±1.1 | 92.7±1.4 | **97.5±1.7** |
| | u-d | 37.3±3.7 | 44.8±11.6 | 83.5±7.0 | 39.8±3.1 | 90.7±2.9 | 88.6±5.7 | 86.5±4.3 | **90.0±1.7** |
| | m-p | 65.8±11.6 | 0.3±0.4 | 89.5±3.4 | 73.6±6.7 | 76.3±2.5 | 82.4±5.8 | 77.9±5.3 | **94.8±3.5** |
| | m-d | 67.3±3.6 | 0.3±0.4 | 83.5±8.2 | 77.2±2.3 | 69.3±3.3 | 80.4±8.9 | 76.3±1.8 | **95.2±2.5** |
| | l-p | 20.8±7.3 | 0.0±0.0 | 52.2±29.0 | 54.2±10.9 | 16.3±11.1 | 20.6±16.3 | 50.9±4.9 | **84.2±5.2** |
| | l-d | 20.5±13.2 | 0.0±0.0 | 64.0±5.4 | 56.6±6.5 | 41.0±10.7 | 45.2±6.9 | 50.6±4.7 | **73.5±6.6** |
| Antmaze | | 304.5±41.3 | 116.2±51.6 | 470.5±54.5 | 397.0±32.2 | 390.3±32.4 | 416.0±44.7 | 434.9±22.4 | **561.0±23.3** |
| pen | human | 13.7±17.0 | -3.9±0.2 | 103.1±8.5 | 47.5±7.4 | 33.7±7.6 | 80.1±16.9 | 93.2±6.9 | **122.2±5.7** |
| | cloned | 1.0±6.6 | 5.1±5.3 | 102.8±7.8 | 58.3±10.4 | 35.7±3.1 | 71.8±35.2 | 88.5±12.3 | **110.1±11.2** |
| | expert | -1.4±2.3 | -1.4±2.3 | 152.1±6.3 | 137.8±5.0 | 130.3±4.2 | 92.6±10.6 | 142.5±7.5 | **152.6±1.2** |
| Adroit | | 13.3±25.9 | -0.2±7.8 | 358±22.6 | 243.6±22.9 | 199.7±14.9 | 244.5±62.7 | 324.2±26.7 | **384.9±18.1** |
| Runtime | | 24.2 | 4.7 | 5.1 | 7.1 | 29.0 | 5.0 | 7.2 | 7.7 |

## 5 EXPERIMENT

We evaluate CAR in terms of performance, computational efficiency, effect of mixed policy, hyperparameter sensitivity and online fine-tuning ability on D4RL benchmark (Fu et al., 2020).

### 5.1 COMPARISONS ON D4RL BENCHMARK

**Tasks.** We evaluate CAR on three diverse D4RL tasks: the standard MuJoCo locomotion benchmarks, the high-dimensional Adroit manipulation tasks, and the sparse-reward Antmaze tasks.

**Baselines.** We compare CAR against a comprehensive set of offline RL baselines. These include: conservative Q-learning methods (CQL (Kumar et al., 2020), RORL (Yang et al., 2022)); traditional policy constraint methods (TD3+BC (Fujimoto & Gu, 2021), reBRAC (Tarasov et al., 2023)); explicit density regularization method (SPOT (Wu et al., 2022)); and refined policy constraint methods (CPI (Ma et al., 2024), iTRPO (Zhang & Tan, 2024)).

**Comparison.** The results presented in Table 1 demonstrate that our method, CAR, achieves SOTA performance across a diverse suite of offline reinforcement learning benchmarks. Its advantage is most pronounced in the challenging sparse-reward Antmaze domain, significantly outperforming the next-best method by a large margin and demonstrates a breakthrough capability in long-horizon trajectory stitching. This success extends to the high-dimensional Adroit tasks, where it sets new records on three tasks. To MuJoCo, CAR significantly outperforms similar methods, especially on low-quality datasets. While RORL performs exceptionally well on MuJoCo, which can be attributed to its ensemble model and generalization on OOD states, CAR still surpasses it on several tasks. Overall, these results highlight the superior performance of CAR.

**Runtime.** We tested the runtime (ms/step) of CAR on hopper-medium-v2 using a GeForce RTX 2080Ti. The results of CAR and other baselines are shown in Table 1. While CAR is 8% slower in

runtime than our baseline, SPOT, it still significantly faster than ensemble model methods. This is because CAR only requires an extra training step of VAE model.

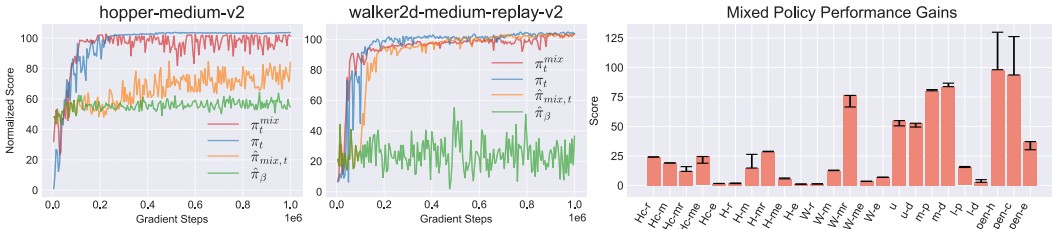

Figure 1: Comparison of policy evaluation methods and mixed policy performance. The figure shows the training curves for different policy on two tasks and the performance gains of the mixed policy over the behavior policy across all tasks.

## 5.2 MIXED POLICY VS LEARNED POLICY

**Choice of Policy Evaluation.** We first test whether the mixed policy $\hat{\pi}_{mix,t}$ should be used to compute the TD target during policy evaluation. Figure 1 presents a comparison of two distinct approaches: a standard learned policy ($\pi_t$) and a policy trained with a mixed-policy TD target ($\pi_t^{mix}$). In the training curve for hopper-medium-v2, we observe that $\pi_t^{mix}$ shows faster initial performance improvement. This is attributed to $\hat{\pi}_{mix,t}$ guaranteeing act within data support. However, its generative nature causes it to assign some probability to suboptimal actions, ultimately leading to a lower final performance than $\pi_t$. Interestingly, this phenomenon is related to the dataset. For walker2d-medium-replay-v2, when the performance of $\hat{\pi}_{mix,t}$ is comparable with $\pi_t$, those two approaches performs similar as well. In summary, to pursue broader applicability across datasets of varying quality, we choose the learned policy for policy evaluation and as the final output policy of our algorithm.

**Mixed Policy Performance Gains.** To evaluate the effectiveness of the mixed policy, we present its performance gains relative to the behavior policy in Figure 1. The height of each bar represents the improvement in score of $\hat{\pi}_{mix,t}$ over $\hat{\pi}_\beta$. We designed the error bars to show changes in performance variance. An upward error bar indicates an increase in variance compared to the behavior policy, while a downward bar indicates a decrease. The mixed policy achieves positive performance gains across all tasks. Furthermore, the variance increases on most tasks, suggesting that the mixed policy's performance is more diverse. This diversity allows it to achieve higher scores, but its generative nature prevents it from completely avoiding suboptimal actions.

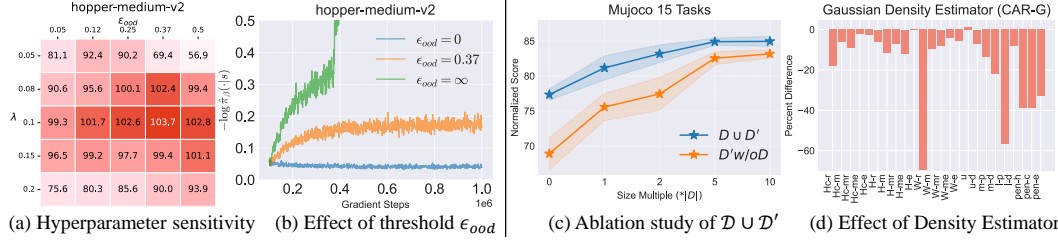

Figure 2: Impact of key design choices in CAR. (a-b) Performance of varying the hyperparameters and effect of threshold $\epsilon_{ood}$. (c-d) Ablation study of auxiliary dataset and effect of density estimator.

## 5.3 HYPERPARAMETER STUDY

**Sensitivity of $\epsilon_{ood}$ and $\lambda$.** CAR involves two key hyperparameters: the KL regularization coefficient $\lambda$ and the support threshold $\epsilon_{ood}$. The sensitivity analysis in Figure 2(a) reveals that $\lambda$ has a more dominant influence on final performance. This is because $\lambda$ directly affect policy optimization, controlling the policy pursues high-value actions, whereas $\epsilon_{ood}$ serves as a final safeguard.

**Effect of $\epsilon_{ood}$.** Figure 2(b) demonstrates how $\epsilon_{ood}$ controls policy behavior. We record the negative log-probability of the learned policy's actions under the behavior policy (as line 11 in Algorithm 1). A well-tuned $\epsilon_{ood}$ achieve balance, enabling the policy to explore low-probability, in-support regions. Conversely, without the threshold ($\epsilon_{ood} = \infty$), the policy is free to progressively drift away from the data support, ultimately leading to OOD actions.

### 5.4 ABLATION STUDY

**Size of $\mathcal{D} \cup \mathcal{D}'$.** We analyze the auxiliary dataset $\mathcal{D}'$, which stores samples from the mixed policy. Figure 2(c) shows the average MuJoCo score under different size $|\mathcal{D}'|$. First, we care about whether it is necessary to retain the offline dataset $\mathcal{D}$. The results ($\mathcal{D} \cup \mathcal{D}'$) reveal that retaining the original offline data is critical, because relying solely on the auxiliary dataset ($\mathcal{D}'$ w/o $\mathcal{D}$) leads to a significant performance drop, as the model forgets the foundational behaviors within $\mathcal{D}$. Next, we observe sensitivity analysis of the size. We find that an auxiliary dataset five times the size of $\mathcal{D}$ offers an effective trade-off between performance gains and computational overhead. Consequently, CAR involves augmenting the original dataset $\mathcal{D}$ with an auxiliary dataset $\mathcal{D}'$ five times its size.

**Density Estimator.** Consistent with previous works (Zhang & Tan, 2024; Mao et al., 2023), we explore replacing the VAE model with a Gaussian density estimator, a variant we term CAR-G. Results in Figure 2(d) shown that the performance of CAR-G is much lower. We attribute this primarily to the complex, multimodal distribution of the mixed dataset $\mathcal{D} \cup \mathcal{D}'$, which originates from multiple historical policies. A Gaussian density estimator, by modeling data as unimodal, may struggle with such complexity and lead to suboptimal actions. In contrast, the VAE, as a generative model, is better equipped to capture these multimodal distributions. In addition, the Diffusion model provides better performance, a result we detail in Appendix B.3.

### 5.5 ONLINE FINE-TUNING AFTER OFFLINE RL

The pluggable design of CAR enables a seamless application of online fine-tuning, which means that we only need to gradually decay the regularization strength $\lambda$ in the online phase in order to avoid overly conservatism. Like baseline SPOT we build upon, we fix the VAE model during online fine-tuning. As the Results shown in Figure 3, the superior offline policy provided by CAR serves as a much stronger starting point for online fine-tuning, allowing to converge to better performance than other methods. We present more results in Appendix B.5.

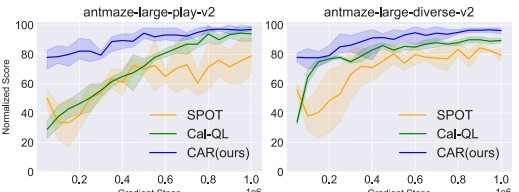

Figure 3: Online fine-tuning results on 2 AntMaze tasks, showing initial performance after offline RL and performance after 1M steps of online RL. (5 seeds)

### 5.6 CONCLUSION

We present **C**overage-**A**ware **R**eference Policy (CAR), which progressively refines behavior regularization by using a tractable propose-and-verify sampling mechanism to augment the static behavior policy. Empirically, CAR obtains excellent performance across various tasks in the D4RL benchmarks while maintaining high computational efficiency, requiring only an additional update step for the density estimator with negligible computational overhead.

**Limitation & Future Work.** One limitation of our method lies in its dependency on the accuracy of the density estimator. As demonstrated by prior work (Zhang et al., 2023), the capacity of the generative model to capture complex, multimodal distributions is crucial for the effectiveness of policy constraint methods. Another limitation is that the practical implementation of CAR leads to a gradual refinement of the behavior regularization. To maintain computational efficiency, the augmentation is archieved with only a single batch of samples per iteration, which constitutes a minor update relative to the large scale of the offline dataset. An exciting direction for future work would be to develop methods for creating a more sample-efficient auxiliary dataset that can effectively approximate our ideal reference policy with fewer samples.

REPRODUCIBILITY STATEMENT

Our source code, including necessary configurations, is provided in the supplementary material. The experimental results can be fully reproduced by directly running the provided scripts.

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

# A PROOFS

This section provides the proofs for the theories in the paper.

## A.1 PROOF OF THEOREM 1

**Theorem 1** (Coverage guarantee) For any $\pi_{mix,t}(a|s)$ in Eq.(6) and let $\epsilon > 0$ be the support threshold, the coverage condition is bounded above for any $t \geq 1$:

$$\max_{s,a} \frac{\pi_{mix,t}(a|s)}{\pi_\beta(a|s)} \leq \frac{1}{\epsilon} \tag{13}$$

*Proof.* We prove Theorem 1 by induction on the iteration step $t$.

Base Case: We check whether Theorem 1 is true for $t = 1$:

$$\forall s, a, \frac{\pi_{mix,1}(a|s)}{\pi_\beta(a|s)} = \tau_1(s)\frac{\hat{\pi}_1(a|s)}{\pi_\beta(a|s)} + 1 - \tau_1(s) \qquad \text{(by Eq. (6))}$$

$$= \frac{\pi_1(a|s)\mathbb{I}(\pi_\beta(a|s) \geq \epsilon)}{\pi_\beta(a|s)} + 1 - \tau_1(s) \qquad \text{(by Eq. (7))}$$

$$= \mathbb{I}(\pi_\beta(a|s) \geq \epsilon)\frac{\pi_1(a|s)}{\pi_\beta(a|s)} + 1 - \tau_1(s) \qquad \text{(by Eq. (7))}$$

$$\leq \frac{\pi_1(a|s)}{\epsilon} - \pi_1(a|s) - \sum_{a'}\mathbb{I}(\pi_\beta(a'|s) \geq \epsilon)\pi_1(a'|s) + 1$$

(The maximum case is when $\pi_\beta(a|s) = \epsilon$)

$$\leq \frac{1}{\epsilon}$$

(The maximum case is when $\pi_1$ only has probability on current $a$)

Inductive Hypothesis: We assume that Theorem 1 is true for $t = k$:

$$\forall s, a, \frac{\pi_{mix,k}(a|s)}{\pi_\beta(a|s)} \leq \frac{1}{\epsilon} \tag{14}$$

Inductive Step: We check whether Theorem 1 is true for $t = k + 1$:

$$\forall s, a, \frac{\pi_{mix,k+1}(a|s)}{\pi_\beta(a|s)} = \tau_{k+1}(s)\frac{\hat{\pi}_{k+1}(a|s)}{\pi_\beta(a|s)} + (1 - \tau_{k+1}(s))\frac{\pi_{mix,k}(a|s)}{\pi_\beta(a|s)} \qquad \text{(by Eq. (6))}$$

$$\leq \tau_{k+1}(s)\frac{\hat{\pi}_{k+1}(a|s)}{\pi_\beta(a|s)} + (1 - \tau_{k+1}(s))\frac{1}{\epsilon} \qquad \text{(by Eq. (7))}$$

$$= \mathbb{I}(\pi_\beta(a|s) \geq \epsilon)\frac{\pi_{k+1}(a|s)}{\pi_\beta(a|s)} + (1 - \tau_{k+1}(s))\frac{1}{\epsilon} \qquad \text{(by Eq. (7))}$$

$$\leq \frac{\pi_{k+1}(a|s)}{\epsilon} - \frac{\pi_{k+1}(a|s)}{\epsilon} - \frac{1}{\epsilon}\sum_{a'}\mathbb{I}(\pi_\beta(a'|s) \geq \epsilon)\pi_{k+1}(a'|s) + \frac{1}{\epsilon}$$

(The maximum case is when $\pi_\beta(a|s) = \epsilon$)

$$\leq \frac{1}{\epsilon}$$

(The maximum case is when $\sum_{a'}\mathbb{I}(\pi_\beta(a'|s) \geq \epsilon)\pi_{k+1}(a'|s) = 0$)

End proof that in the maximal case Theorem 1 is true for every $t \geq 1$. $\square$

## A.2 Proof of Theorem 2

We first introduced the necessary assumptions and lemmas that used in our proofs.

Offline RL addresses the problem of learning an optimal policy from a previously collected dataset, denoted as $\mathcal{D} = \{s_i, a_i, r_i, s_i'\}_{i=1}^n$. The experiences in this dataset are typically assumed to be generated by an unknown behavior policy $\pi_\beta$ acting in the environment: $(s_i, a_i) \sim \mu, s_i' \sim P(\cdot|s_i, a_i), r_i = R(s_i, a_i)$, where $\mu$ represents the state-action occupancy. In large and complex state-action spaces, function approximation is typically employed to approximate the Q-function. It often search for a good policy with the help of a parameterized function approximators class $\mathcal{F} \subset (\mathcal{S} \times \mathcal{A} \rightarrow [0, V_{\max}])$ to model $Q^\pi$, where $V_{\max} = R_{\max}/(1 - \gamma)$. We now recall two standard expressivity assumptions on $\mathcal{F}$.

**Assumption 1** (Realizability). *For any $\pi \in \Pi$, we have*

$$\inf_{f \in \mathcal{F}} \sup_{admissible\ \nu} \|f - \mathcal{T}^\pi f\|_{2,\nu}^2 \le \varepsilon_\mathcal{F},$$

where $\|f\|_{2,\mathcal{D}}^2 = \mathbb{E}_\mathcal{D}[f^2]$, an admissible distribution $\nu$ means that the data of any possible policy and the behavior policy $\nu \in \{d_{\pi'} : \pi' \in \Pi\} \cup \{\mu\}$. Assumption 1 requires that for every $\pi \in \Pi$, there exists $f \in \mathcal{F}$ that well-approximates $Q^\pi$.

**Assumption 2** (Completeness). *For any $\pi \in \Pi$, we have*

$$\sup_{f \in \mathcal{F}} \inf_{f' \in \mathcal{F}} \|f' - \mathcal{T}^\pi f\|_{2,\mu}^2 \le \varepsilon_{\mathcal{F},\mathcal{F}}.$$

Assumption 2 asserts that $\mathcal{F}$ is approximately closed under $\mathcal{T}^\pi$. Such assumptions are widely used in RL theory (Chen & Jiang, 2019).

In the following lemma, we show that for any $\pi_{mix,t}$ in Eq.(6) and any $\epsilon > 0$, apply Bellman Residual (Error) Minimization (Xie et al., 2021) with function approximation deals with the following regression problem on dataset:

$$f_t = \arg\min_{f \in \mathcal{F}} \mathcal{E}(f, \pi_{mix,t}; \mathcal{D}), \tag{15}$$

where, $\mathcal{E}(f, \pi_{mix,t}; \mathcal{D}) := \|f(s,a) - r - \gamma f(s', \pi_{mix,t})\|_{2,\mathcal{D}}^2 - \min_{f' \in \mathcal{F}} \|f'(s,a) - r - \gamma f(s', \pi_{mix,t})\|_{2,\mathcal{D}}^2$, the estimated function $f_t$ is actually the true Q-value of $\pi_{mix,t}$ in a specific MDP $\mathcal{M}_t$, denoted by $Q_{\mathcal{M}_t}^{\pi_{mix,t}}$, where dynamic of $\mathcal{M}_t$ is same as the ground-truth MDP $\mathcal{M}$ and the difference between the reward functions of $\mathcal{M}$ and $\mathcal{M}_t$ can be controlled.

Our results depends on the following errors (Theorem A.1 and A.2 in Xie et al. (2021)):

1. Empirical Bellman Error $\varepsilon_r$—For any $\pi_{mix,t}$, if $f_{\pi_{mix,t}}$ is the best estimation of $Q^{\pi_{mix,t}}$ in $\mathcal{F}$ for general function approximation, then $\mathcal{E}(f_{\pi_{mix,t}}; \pi_{mix,t}; \mathcal{D}) \le \varepsilon_r$.

2. Mean-Squared Bellman Error $\varepsilon_b$—For any function $f \in \mathcal{F}$ and any $\mathcal{E}(f_{\pi_{mix,t}}; \pi_{mix,t}; \mathcal{D}) \le \varepsilon_r$, then $\|f - \mathcal{T}^\pi f\|_{2,\mu}^2 \le \varepsilon_b$.

Following lemmas provides a more detailed explanation.

**Lemma 1.** *For any $\pi \in \Pi$, let $f_\pi$ be defined as follows,*

$$f_\pi := \arg\min_{f \in \mathcal{F}} \sup_{admissible\ \nu} \|f - \mathcal{T}^\pi f\|_{2,\nu}^2.$$

*Then, for $\mathcal{E}(f_\pi; \pi; \mathcal{D})$ (defined in Eq.(3.1)), we have*

$$\mathcal{E}(f_\pi; \pi; \mathcal{D}) \le \frac{139 V_{\max}^2}{n} \log \frac{|\mathcal{F}||\Pi|}{\delta} + 39\varepsilon_\mathcal{F} =: \varepsilon_r.$$

Next lemma show that $\mathcal{E}(f, \pi; \mathcal{D})$ could effectively estimate $\|f - \mathcal{T}^\pi f\|_{2,\mu}^2$.

**Lemma 2.** *For any $\pi \in \Pi$, $f \in \mathcal{F}$, and any $\varepsilon_r > 0$, if $\mathcal{E}(f; \pi; \mathcal{D}) \le \varepsilon_r$, then,*

$$\|f - \mathcal{T}^\pi f\|_{2,\mu}^2 \le V_{\max} \sqrt{\frac{231 \log |\mathcal{F}||\Pi|/\delta}{n}} + \sqrt{\varepsilon_{\mathcal{F},\mathcal{F}}} + \sqrt{\varepsilon_{\mathcal{F},\mathcal{F}} + \varepsilon_r} =: \sqrt{\varepsilon_b}$$

Next lemma shows the difference between the reward functions of $\mathcal{M}$ and $\mathcal{M}_t$ can be controlled.

**Lemma 3.** *Let $f_t$ satisfies $\mathcal{E}(f_t, \pi_{mix,t}, \mathcal{D}) \leq \varepsilon_r$ for any $\pi_t$ in RL. Then, there exists an MDP $\mathcal{M}_t = (\mathcal{R}_t)$, where the other elements of $\mathcal{M}_t$ are same as the environment MDP $\mathcal{M}$. We have $\|R_t(s,a) - R(s,a)\|_{2,\mu}^2 \leq \varepsilon_b$, such that $f_t = Q_{\mathcal{M}_t}^{\pi_t}$.*

*Proof.* We use the definition of the bellmanE expectation operator (Eq. (1)) to define the reward $R_t$:

$$R_t(s,a) := f_t(s,a) - \gamma \mathbb{E}_{s' \sim P(\cdot|s,a)} [f_t(s', \pi_t)]. \tag{16}$$

for $\|R_t(s,a) - R(s,a)\|_{2,\mu}$, we have

$$\|R_t(s,a) - R(s,a)\|_{2,\mu}^2 = \left\| f_t(s,a) - \gamma \mathbb{E}_{s' \sim P(\cdot|s,a)} [f_t(s', \pi_t)] - R(s,a) \right\|_{2,\mu}^2$$

$$= \left\| f_t(s,a) - \left( R(s,a) + \gamma \mathbb{E}_{s' \sim P(\cdot|s,a)} [f_t(s', \pi_t)] \right) \right\|_{2,\mu}^2$$

$$= \| f_t - \mathcal{T}^{\pi_t} f_t \|_{2,\mu}^2$$

$\square$

Then we rely on theorem 1 to provide two useful inferences.

**Corollary 1.** *Under Theorem 1, for any $\pi_{mix,t}$ in Eq.(6) and any $\epsilon > 0$, the KL divergence between the mixed policy and the behavior policy is bound above:*

$$D_{KL}(\pi_{mix,t} \| \pi_\beta) \leq -\log \epsilon \tag{17}$$

*Proof.*

$$D_{KL}(\pi_{mix,t} \| \pi_\beta) = D_{KL}(\tau_t(s)\hat{\pi}_t + (1 - \tau_t(s))\pi_{mix,t-1} \| \pi_\beta)$$

$$\leq \tau_t(s) D_{KL}(\hat{\pi}_t \| \pi_\beta) + (1 - \tau_t(s)) D_{KL}(\pi_{mix,t-1} \| \pi_\beta)$$

$$= \tau_t(s) \sum_a \hat{\pi}_t(a|s) \log \frac{\hat{\pi}_t(a|s)}{\pi_\beta(a|s)} + (1 - \tau_t(s)) \sum_a \hat{\pi}_{mix,t-1}(a|s) \log \frac{\hat{\pi}_{mix,t-1}(a|s)}{\pi_\beta(a|s)}$$

$$\leq \tau_t(s) \sum_a \hat{\pi}_t(a|s) \log \frac{\hat{\pi}_t(a|s)}{\pi_\beta(a|s)} + (1 - \tau_t(s)) \sum_a \hat{\pi}_{mix,t-1}(a|s) \log \frac{1}{\epsilon}$$

$$= \tau_t(s) \sum_{a:\pi_\beta(a|s) \geq \epsilon} \pi_t(a|s) \log \frac{\pi_t(a|s)}{\pi_\beta(a|s)} + (1 - \tau_t(s)) \log \frac{1}{\epsilon}$$

$$\leq \tau_t(s) \sum_{a:\pi_\beta(a|s) \geq \epsilon} \pi_t(a|s) \log \frac{\pi_t(a|s)}{\epsilon} + (1 - \tau_t(s)) \log \frac{1}{\epsilon}$$

(The maximum case is when $\pi_\beta(a|s) = \epsilon$)

$$\leq \tau_t(s) \sum_{a:\pi_\beta(a|s) \geq \epsilon} \pi_t(a|s) \log \frac{\pi_t(a|s)}{\epsilon} + (1 - \tau_t(s)) \log \frac{1}{\epsilon}$$

$$\leq -\log \epsilon$$

$\square$

**Corollary 2.**

$$\max_{s,a} \frac{d^{\pi_{mix,t}}(s,a)}{d^{\pi_\beta}(s,a)} \leq \frac{|\mathcal{S}|}{\epsilon} \tag{18}$$

*Proof.* We define the discounted state-action occupancy : $d^\pi(s,a) = (1-\gamma)\sum_{t=0}^{\infty}\gamma^t Pr(s_t = s, a_t = a; \pi, \rho_0)$. Then, we can obtain a recursive expression for any $s, a$:

$$d^{\pi_{mix,t}}(s,a) = (1-\gamma)\rho_0(s)\pi_{mix,t}(a|s) + \gamma\sum_{s',a'}d^{\pi_{mix,t}}(s',a')P(s|s',a')\pi_{mix,t}(a|s)$$

$$\leq (1-\gamma)\rho_0(s)\frac{\pi_\beta(a|s)}{\epsilon} + \gamma\sum_{s',a'}d^{\pi_{mix,t}}(s',a')P(s|s',a')\frac{\pi_\beta(a|s)}{\epsilon} \quad \text{(By Theorem 1)}$$

$$= \frac{\pi_\beta(a|s)}{\epsilon}\left[(1-\gamma)\rho_0(s) + \gamma\sum_{s',a'}d^{\pi_{mix,t}}(s',a')P(s|s',a')\right]$$

$$= \frac{\pi_\beta(a|s)}{\epsilon}d^{\pi_{mix,t}}(s) \qquad\qquad \text{(By definition of state occupancy)}$$

$$\overset{(a)}{\leq} \frac{\pi_\beta(a|s)}{\epsilon}|\mathcal{S}|d^{\pi_\beta}(s)$$

$$= \frac{|\mathcal{S}|}{\epsilon}d^{\pi_\beta}(s,a)$$

Tthe inequality (a) relies on [Chen & Jiang (2019), Proposition 10] stated as: $\frac{d^\pi(s)}{d^{\pi_\beta}(s)} \leq |\mathcal{S}|$. $\qquad\square$

**Theorem 2** Under Theorem 1 and approximate completeness assumption, for any $\pi_{mix,t}$ in Eq.(6) and any $\epsilon > 0$, apply Bellman Residual (Error) Minimization (Xie et al., 2021) and then with probability at least $1-\delta$ the following bound holds:

$$J(\pi_\beta) - J(\pi_{mix,t}) \leq \frac{V_{\max}\sqrt{-\log\epsilon}}{\sqrt{2}(1-\gamma)} + (1+\frac{|\mathcal{S}|}{\epsilon})\frac{\sqrt{\epsilon_b}}{1-\gamma}, \tag{19}$$

where $\sqrt{\epsilon_b} \leq \mathcal{O}\left(V_{\max}\sqrt{\frac{\log|\mathcal{F}|/\delta}{n}}\right)$ is mean-squared bellman error.

*Proof.*

$$J(\pi_\beta) - J(\pi_{mix,t}) =$$
$$\underbrace{[J_M(\pi_\beta) - J_{M_t}(\pi_\beta)]}_{\text{Term 1}} + \underbrace{[J_{M_t}(\pi_\beta) - J_{M_t}(\pi_{mix,t})]}_{\text{Term 2}} + \underbrace{[J_{M_t}(\pi_{mix,t}) - J_M(\pi_{mix,t})]}_{\text{Term 3}}$$

For term 1 and 3, from the definition of the Bellman equation, we can obtain:

$$J(\pi) - J_{\mathcal{M}_t}(\pi) = \frac{1}{1-\gamma}E_{d^\pi}[R(s,a) - R_t(s,a)]$$

$$= \frac{1}{1-\gamma}E_{d^\pi}[\mathcal{T}^{\pi_t}f_t(s,a) - f_t(s,a)] \qquad \text{(By lemma 3)}$$

$$= -\frac{1}{1-\gamma}E_{d^\pi}[f_t(s,a) - \mathcal{T}^{\pi_t}f_t(s,a)]$$

Then, we have:

$$|J(\pi_\beta) - J_{M_t}(\pi_\beta)| = \frac{1}{1-\gamma}|E_\mu[f_t(s,a) - \mathcal{T}^{\pi_t}f_t(s,a)]|$$

$$\leq \frac{1}{1-\gamma}\sqrt{E_\mu[(f_t(s,a) - \mathcal{T}^{\pi_t}f_t(s,a))^2]}$$

(By Cauchy-Schwarz inequality for random variables $|\mathbb{E}[XY]| \leq \sqrt{\mathbb{E}[X^2]\mathbb{E}[Y^2]}$)

$$= \frac{1}{1-\gamma}\|f_t - \mathcal{T}^{\pi_t}f_t\|_{2,\mu}$$

$$\leq \frac{\sqrt{\epsilon_b}}{1-\gamma}$$

$$|J(\pi_{mix,t}) - J_{M_t}(\pi_{mix,t})| \leq \frac{1}{1-\gamma}\|f_t - \mathcal{T}^{\pi_t} f_t\|_{2,d_{\pi_{mix,t}}}$$

$$= \frac{1}{1-\gamma}\frac{d_{\pi_{mix,t}}(s,a)}{\mu(s,a)}\|f_t - \mathcal{T}^{\pi_t} f_t\|_{2,\mu}$$

$$\leq \frac{|\mathcal{S}|\sqrt{\varepsilon_b}}{\epsilon(1-\gamma)} \qquad \text{(By corollary 2)}$$

For term 2, using the performance difference lemma (Kakade & Langford, 2002), we have

$$J_{\mathcal{M}_t}(\pi_\beta) - J_{\mathcal{M}_t}(\pi_{mix,t}) = \frac{1}{1-\gamma}\mathbb{E}_{s\sim d_{\pi_\beta,\mathcal{M}_t}}\left[Q^{\pi_{mix,t}}_{\mathcal{M}_t}(s,\pi_\beta) - Q^{\pi_{mix,t}}_{\mathcal{M}_t}(s,\pi_{mix,t})\right]$$

$$\stackrel{(a)}{=} \frac{1}{1-\gamma}\sum_s d_{\pi_\beta}(s)\sum_a[(\pi_\beta(a|s) - \pi_{mix,t}(a|s))f_t(s,a)]$$

$$\leq \frac{1}{1-\gamma}\sum_s d_{\pi_\beta}(s)\sum_a[|\pi_\beta(a|s) - \pi_{mix,t}(a|s)||f_t(s,a)|]$$

$$\leq \frac{V_{\max}}{1-\gamma}\sum_s d_{\pi_\beta}(s)\sum_a[|\pi_\beta(a|s) - \pi_t(a|s)|$$

$$= \frac{V_{\max}}{1-\gamma}\sum_s d_{\pi_\beta}(s)\mathrm{D}_{\mathrm{TV}}(\pi_{mix,t}\|\pi_\beta)(s)$$

$$\leq \frac{V_{\max}}{\sqrt{2}(1-\gamma)}\sum_s d_{\pi_\beta}(s)\sqrt{\mathrm{D}_{\mathrm{KL}}(\pi_{mix,t}\|\pi_\beta)(s)} \quad \text{(Pinsker's inequality)}$$

$$\leq \frac{V_{\max}\sqrt{-\log\epsilon}}{\sqrt{2}(1-\gamma)} \qquad \text{(By Corollary 1)}$$

the equation (a) depends on lemma 3, we know the dynamics of $\mathcal{M}_t$ are same as that of the true environment MDP $\mathcal{M}$. Let $d_{\pi_\beta} = d_{\pi_\beta,\mathcal{M}_t}(s)$.

Based on the above 3 terms, we have:

$$J(\pi_\beta) - J(\pi_{mix,t}) = \frac{V_{\max}\sqrt{-\log\epsilon}}{\sqrt{2}(1-\gamma)} + \frac{\sqrt{\epsilon_b}}{1-\gamma} + \frac{|\mathcal{S}|\sqrt{\varepsilon_b}}{\epsilon(1-\gamma)}$$

We completes the proof by applying lemma 2, mean-squared bellman error $\sqrt{\epsilon_b} \leq \mathcal{O}\left(V_{\max}\sqrt{\frac{\log|\mathcal{F}||\Pi|/\delta}{n}}\right)$. $\qquad\square$

# B  IMPLEMENTATION DETAILS AND EXTENDED RESULTS

## B.1  BENCHMARK EXPERIMENTS

**Data.**  We use the datasets from the D4RL benchmark (Fu et al., 2020), of the latest versions, which are v2 for MuJoCo, AntMaze and v1 for Adroit. We provide an explanation of the abbreviations used in the results table. Mujoco Tasks: r = random, m = medium, m-r = medium-replay, m-e = medium-expert, e = expert. Antmaze Tasks: u = umaze, u-d = umaze-diverse, m-p = medium-play, m-d = medium-diverse, l-p = large-play, l-d = umaze-diverse.

**Baselines.**  We report the baseline result in their paper, missing experiment we reproduce the code:

- SPOT (Wu et al., 2022): `https://github.com/thuml/SPOT`
- CORL (Tarasov et al., 2023): `https://github.com/tinkoff-ai/CORL`
- RORL (Yang et al., 2022): `https://github.com/YangRui2015/RORL`
- CPI (Ma et al., 2024): `https://github.com/mamengyiyi/CPI`

**Implementation details.** Our algorithm CAR is based on the implementation of SPOT (Wu et al., 2022). We normalize the states in the dataset for MuJoCo and Adroit but do not normalize the states for AntMaze. Following SPOT, we subtract 1 from rewards for the AntMaze datasets. Hyperparameters used are presented in Table 2. Our code is included in the supplementary material, and the detailed configuration for each task is included within it.

**Hyperparameter tuning.** CAR involves two key hyperparameters: the KL regularization coefficient $\lambda$ and the support threshold $\epsilon_{ood}$. As analyzed in Section 5.3, $\lambda$ has a more dominant influence on the final performance. For hyperparameter tuning, we first determine $\lambda$ based on the baseline method by setting $\epsilon_{ood} = 0$ to achieve the best possible performance. Then, we gradually increase $\epsilon_{ood}$ to obtain further performance improvements. We show more parameter sensitivity analysis in Figure 4.

Table 2: Hyperparameters in CAR

|  | Hyperparameter | Value |
|---|---|---|
|  | Optimizer | Adam |
|  | Critic learning rate | $1 \times 10^{-3}$ |
|  | Actor learning rate | $1 \times 10^{-3}$ |
|  | Batch size | 256 |
|  | Discount factor | 0.99 |
|  | Gradient Steps | $1 \times 10^{6}$ |
| RL training | Warm-up Steps | $1 \times 10^{5}$ |
|  | VAE learning rate | $5 \times 10^{-6}$ |
|  | Target network update rate | 0.005 |
|  | Policy update frequency | 2 |
|  | Number of Critics | 4 (including 2 target network) |
|  | Threshold $\epsilon_{ood}$ | $[0.02, 0.8]$ |
|  | Balance coefficient $\lambda$ | $[0.05, 38]$ |
|  | Dataset size $|\mathcal{D}'|$ | $5 \times |\mathcal{D}|$ |
|  | Actor | input-256-256-output |
| Architecture | Critic | input-256-256-1 |
|  | VAE | input-750-750-2×action dim-750-750-output |

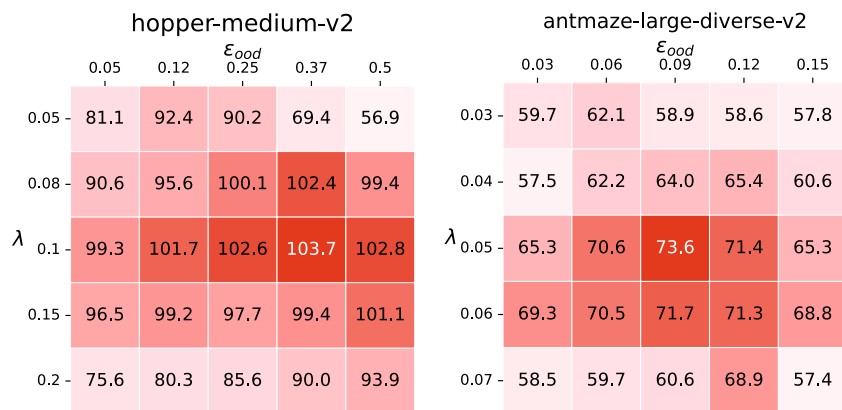

Figure 4: Performance of varying the hyperparameters, which is a expansion of Figure 2(a)

**Learning curves.** Figure 5 and Figure 6 presents the training curves for all tasks in the benchmark experiments. The shaded area indicates the minimum and maximum score across 5 seeds, with the solid line depicting their mean.

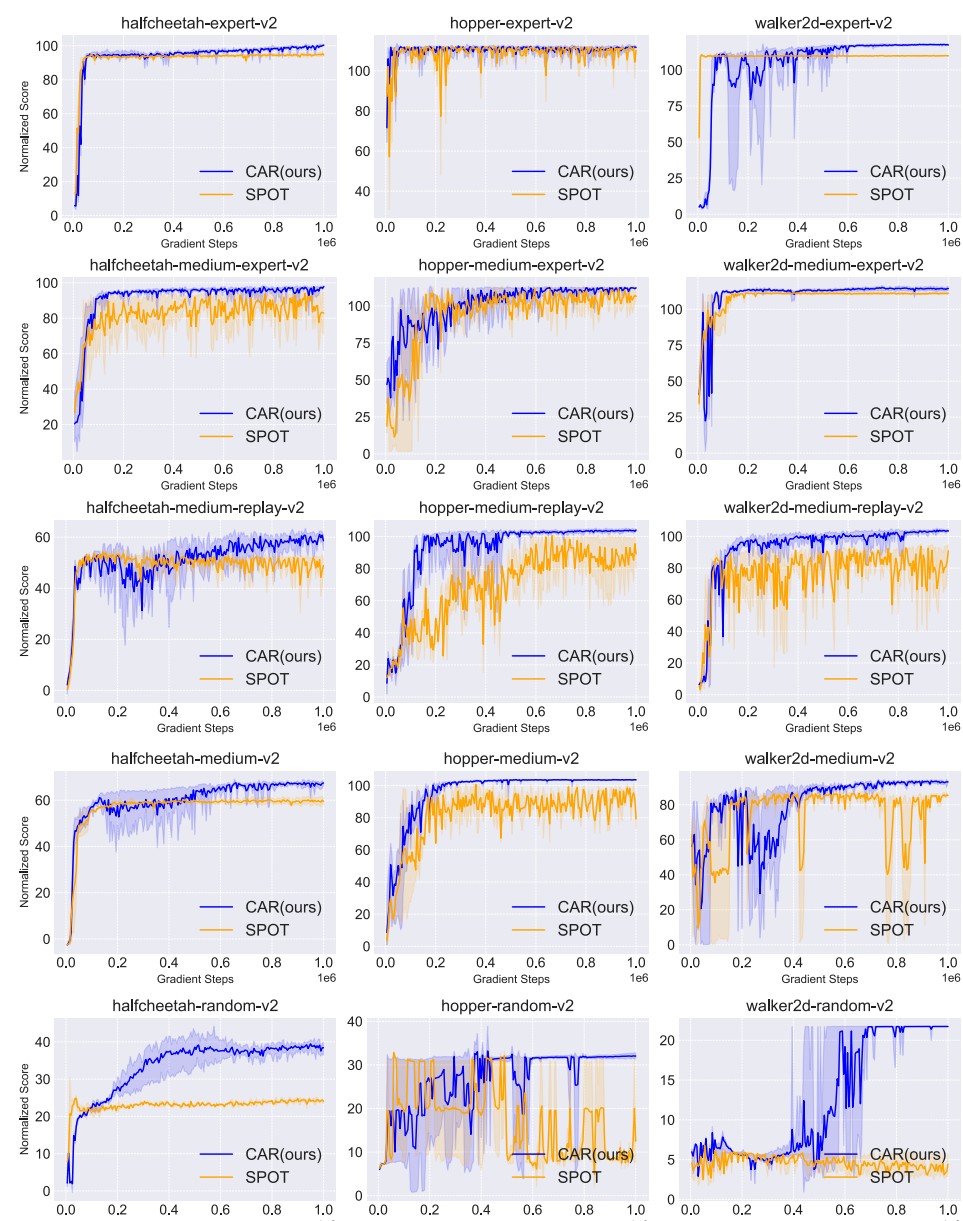

Figure 5: The results on the MuJoCo task compaired with baseline. The shaded area indicates the minimum and maximum score across 5 seeds, with the solid line depicting their mean.

## B.2 MIXED POLICY PERFORMANCE

We present the complete performance change of the mixed policy relative to the behavior policy across all tasks after offline RL training in Table 3.

## B.3 ABLATION STUDY

**Ablation study with variants of CAR.** We consider several variants of CAR to perform an ablation study over the components in our method:

- $\mathcal{D} \cup \mathcal{D}'$: To avoid adding extra computational overhead, CAR is designed to use data that is produced by RL training process. All actions taken by the learned policy $\pi_t$ are essential for updating the Q-value function, rather than being separately outputted by another policy

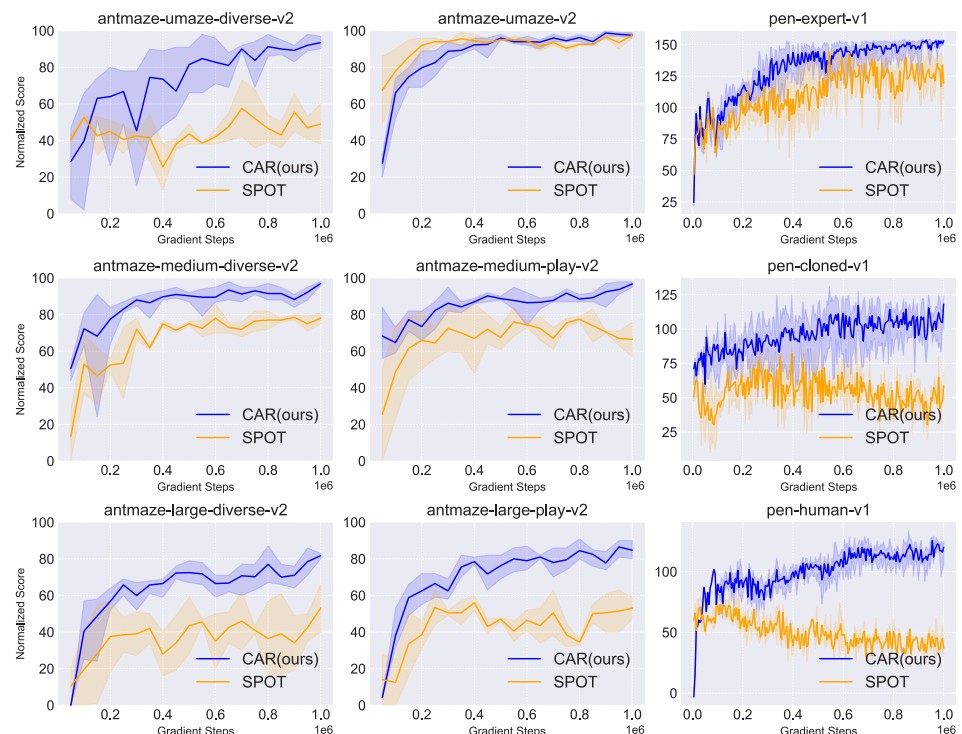

Figure 6: The results on the Antmaze and Adroit task compaired with baseline. The shaded area indicates the minimum and maximum score across 5 seeds, with the solid line depicting their mean.

network. The only additional computational cost comes from two sources: using the VAE to estimate the probability of the behavioral policy with new data and a single VAE model update step. Furthermore, $\mathcal{D}'$ only contains $(s, a)$ data and is stored in CPU memory, so it doesn't increase GPU memory usage.

- Gaussian: We train a parametric model $\hat{\pi}_{mix,t}$, which fits a tanh-Gaussian distribution to the actions $a$ given the states $s$: $\hat{\pi}_{mix,t}(\cdot|s) = \tanh \mathcal{N}(\mu(\cdot|s), \sigma(\cdot|s))$. Then we use $\hat{\pi}_{mix,t}$ as our density estimator in the actor learning objective.

- Diffusion: In the density estimator ablation experiment, we use the Diffusion model instead of VAE. Code of Diffusion model is based on the implementation of OSC (Gao et al., 2025): https://github.com/MoreanP/OSC. Diffusion has performance improvements over VAE in a few tasks, results are presented in Figure 7 (right).

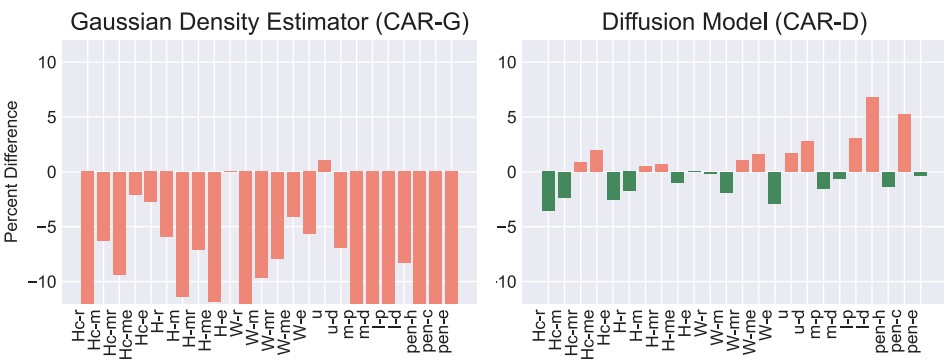

Figure 7: Percent difference of the performance of ablations of density estimator.

Table 3: The mean and standard deviation of averaged normalized scores of VAE behavior policy model and VAE mixed policy model after 1M steps of offline RL training, averaged over 5 seeds and the final 10 evaluations (final 4 for Antmaze).

| | | $\hat{\pi}_\beta$ | $\hat{\pi}_{mix,t}$ |
|---|---|---|---|
| halfcheetah | r | 2.3±0 | 26.3±0.2 |
| | m | 42.4±0.5 | 61.5±0.6 |
| | m-r | 35.6±1.4 | 47.6±5.4 |
| | m-e | 61.1±10.4 | 85.7±4.7 |
| | e | 92.1±0.4 | 93.7±0.5 |
| hopper | r | 2.1±0.6 | 4.2±0.1 |
| | m | 54.5±3.5 | 59.3±15.1 |
| | m-r | 44.5±13.2 | 73.1±13.6 |
| | m-e | 49.5±4.6 | 55.2±5.3 |
| | e | 110.6±0.3 | 111.6±0.7 |
| walker2d | r | 0±0.3 | 1.4±0.2 |
| | m | 71.1±3 | 83.7±3.4 |
| | m-r | 26.1±11 | 102.4±1.2 |
| | m-e | 108.7±0.1 | 112.1±0.3 |
| | e | 108.1±0.2 | 115.1±0.1 |
| antmaze | u | 42.5±5.6 | 97.5±1.1 |
| | u-d | 39.5±4.7 | 92.3±1.5 |
| | m-p | 0.5±0.5 | 80.8±1.5 |
| | m-d | 0.3±0.4 | 84±3.4 |
| | l-p | 0±0 | 15.3±0.8 |
| | l-d | 0±0 | 2.8±2.2 |
| pen | human | 0±0.6 | 98.1±32.3 |
| | cloned | 0±2.7 | 93.7±35.2 |
| | expert | 107.6±20.5 | 144.8±13.6 |

## B.4 COMPUTATION COST

Since CAR requires updates VAE model during training, it introduces extra computational overhead. Table 4 shows a comparison of the computational costs for various baseline methods and CAR on the hopper-medium-v2 task.

Table 4: The runtime of CAR on hopper-medium-v2 using a GeForce RTX 2080Ti GPU and Intel Xeon Gold 5218 CPU.

| | CQL | TD3+BC | reBRAC | SPOT | RORL | CPI | iTRPO | CAR(ours) |
|---|---|---|---|---|---|---|---|---|
| Runtime (ms/step) | 24.2 | 4.7 | 5.1 | 7.1 | 29.0 | 5.0 | 7.2 | 7.7 |
| CPU memory (GB) | 2.5 | 2.5 | 2.5 | 2.5 | 3.6 | 2.5 | 2.5 | 2.8 |
| GPU memory (MB) | 1174 | 1056 | 1056 | 1207 | 2338 | 1172 | 1172 | 1235 |

## B.5 ONLINE FINE-TUNING

**Implementation details.** Following the offline RL phase, we perform online fine-tuning of our models for 1 million gradient steps. During this online phase, we actively gather new data from the environment using an exploration noise of 0.1 and incorporate this data into the replay buffer. We apply a linear decay to the regularization coefficient $\lambda$ throughout the online phase. We conduct experiments on complex AntMaze domains, which feature high-dimensional state and action spaces along with sparse rewards, bootstrapping error can become significant. Consequently, we halt the decay of $\lambda$ when it reaches 20% of its initial value at the 0.2 millionth step. All other training parameters remain consistent the baseline method.

**Comparisons results.** Table 5 presents the results of the final performance comparison, and Figure 8 presents the learning curves.

Table 5: Online fine-tuning results on AntMaze tasks, showing initial performance after offline RL and performance after 1M steps of online RL. All results are averaged over 5 seeds.

| AntMaze | SPOT | Cal-QL | CAR (Ours) |
|---|---|---|---|
| u | $95.0 \rightarrow 99.2 \pm 0.8$ | $76.8 \rightarrow \textbf{99.8} \pm 0.4$ | $99.2 \rightarrow 99.6 \pm 0.4$ |
| u-d | $31.5 \rightarrow 92.6 \pm 2.6$ | $32.0 \rightarrow 98.5 \pm 1.1$ | $95.6 \rightarrow \textbf{99.4} \pm 0.7$ |
| m-p | $76.5 \rightarrow 94.3 \pm 2.2$ | $71.8 \rightarrow 98.8 \pm 1.6$ | $93.6 \rightarrow \textbf{99.0} \pm 0.8$ |
| m-d | $75.0 \rightarrow 93.2 \pm 1.3$ | $62.0 \rightarrow 98.3 \pm 1.5$ | $89.8 \rightarrow \textbf{99.1} \pm 0.7$ |
| l-p | $53.0 \rightarrow 75.5 \pm 3.2$ | $31.8 \rightarrow \textbf{97.3} \pm 1.8$ | $78.0 \rightarrow 96.9 \pm 1.1$ |
| l-d | $56.0 \rightarrow 80.8 \pm 3.5$ | $44.0 \rightarrow 91.5 \pm 3.9$ | $77.8 \rightarrow \textbf{96.5} \pm 1.2$ |
| **Average** | $64.5 \rightarrow 89.2$ | $53.04 \rightarrow 97.33$ | $89.0 \rightarrow \textbf{98.4}$ |

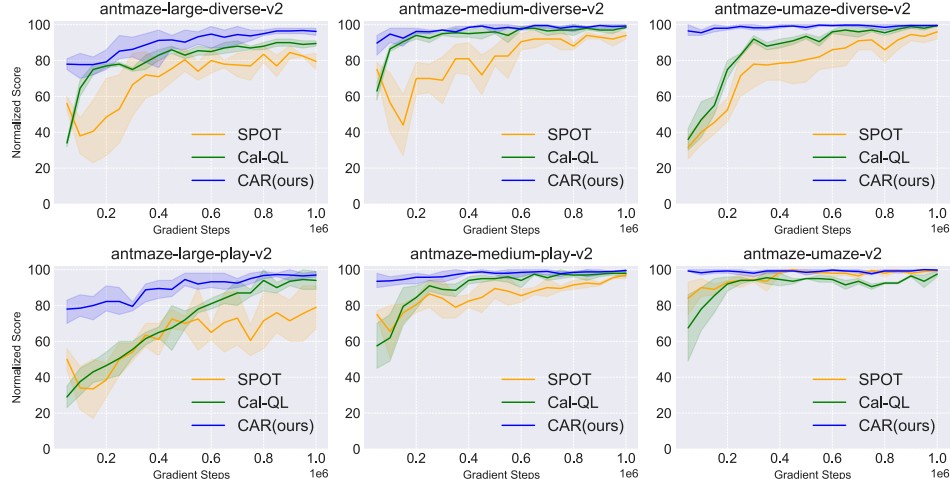

Figure 8: Online fine-tuning training curves on AntMaze tasks, showing initial performance after offline RL and performance after 1M steps of online RL. (5 seeds)

## C  THE USE OF LARGE LANGUAGE MODELS

We utilize large language models (LLMs) to polish the writing and correct spelling.

