# OpenReview forum: "Decoupling Policy Improvement and Conservatism in Offline Reinforcement Learning"
_ICLR.cc/2026/Conference — Submitted to ICLR 2026_

### Official Review · Reviewer_x78z · 2025-10-26

**Soundness:** 2
**Presentation:** 3
**Contribution:** 2
**Rating:** 2
**Confidence:** 4

**Summary:**

This paper introduces the Coverage-Aware Reference Policy (CAR), a novel method for offline reinforcement learning that aims to resolve the conflict between policy improvement and conservatism. The core contribution is the principle of decoupling these two objectives via a "propose-and-verify" mechanism, where the learned policy proposes actions that are then verified for data support before augmenting a reference policy. The method is supported by theoretical guarantees and demonstrates strong, state-of-the-art empirical results across a wide range of D4RL benchmarks. The problem framing is insightful, and the empirical validation is thorough.

**Strengths:**

1. Insightful Problem Framing: The paper compellingly frames a core offline RL challenge as a harmful coupling of policy improvement and conservatism, offering an elegant "propose-and-verify" mechanism to decouple these objectives.

2. State-of-the-Art Empirical Results: CAR achieves outstanding performance across diverse D4RL benchmarks, significantly outperforming strong baselines, particularly in challenging sparse-reward Antmaze tasks.

3. Paper is well-written and easy to follow.

**Weaknesses:**

1. Lacks some related work in the in-sample learning paradigm in offline RL, e.g., IQL, SQL. Please enhance this direction of research in offline RL.
2. The authors empirically justify this choice in Section 5.2, showing that using $\pi_t$ leads to higher final performance. While the results are strong, this creates a disconnect between the theory and the final algorithm. A deeper discussion on why this theoretically riskier approach works better in practice would significantly strengthen the paper's contribution.
3. Potential Bias in Hyperparameter Tuning Strategy: The two-stage hyperparameter tuning strategy described for CAR may create an unfair advantage. First optimizing the regularization coefficient $\lambda$ and then separately tuning the support threshold $\epsilon_{ood}$ could result in a more exhaustive search for CAR compared to its baselines. The authors should clarify if an equivalent tuning budget was used for all methods to ensure the performance gains are attributable to the method itself and not the tuning process.

**Questions:**

1. A question regarding Section 4.2 and Equation (5):
The mixing coefficient $\tau_t(s)$ is defined in Equation (5) as an expectation, $\\mathbb{E}_{a \sim \pi_t}[\mathbb{I}(\ldots)]$, which implies it is a continuous probability value, $P\in [0,1]$. However, the subsequent procedural description ("...proposes an action... If the action is accepted...") describes a single sampling event which results in a binary outcome (accept or reject). To clarify: should we interpret $\tau_t(s)$ as the continuous probability of this acceptance event, and the described procedure as a single Bernoulli trial governed by that probability? Or is there an alternative implementation of the expectation in practice?

2. Following question 1, in Algorithm 1 (Lines 11-13), I think $\tau_t(s)$ is a binary outcome with a single sampling event. Could the authors please clarify how the algorithm ensures stability and avoids sample bias with this single-sample approach? Alternatively, if multiple samples are used to form a more stable estimate, could you comment on the associated computational overhead?

3. Regarding the proof of **Theorem 1** in Appendix A.1:
3.1 The proof contains algebraic steps that are difficult to follow and may contain typos, particularly in the inequality expansions. Additionally, the derivation appears to implicitly use the identity $\tau_t(s) = Z_t(s)$ without explicitly stating it, creating a logical leap. Also, this clarification is highly related to Q1 and Q2.

3.2 Could the authors please revise the proof to clarify these steps for improved readability and rigor?
$\begin{aligned} & =\mathbb{I}\left(\pi_\beta(a \mid s) \geq \epsilon\right) \frac{\pi_1(a \mid s)}{\pi_\beta(a \mid s)}+1-\tau_1(s) \\ & \leq \frac{\pi_1(a \mid s)}{\epsilon}-\pi_1(a \mid s)-\sum_{a^{\prime}} \mathbb{I}\left(\pi_\beta\left(a^{\prime} \mid s\right) \geq \epsilon\right) \pi_1\left(a^{\prime} \mid s\right)+1\end{aligned}$

4. Regarding the construction of the auxiliary dataset $\mathcal{D}'$: The paper employs a progressive,
append-only approach, which is noted as a "minor update" per iteration. Have the authors explored more adaptive strategies for managing $\mathcal{D}'$? For instance, have you considered periodically rebuilding the dataset to better reflect the current policy, or using a prioritized sampling scheme to manage its contents more effectively? An ablation study on these alternatives would be valuable to justify the current design choice.

---

> ### Author Response · Authors · 2025-11-25
>
> We sincerely thank you for your concerns and the effort you have put into our work. We will carefully correct the spelling mistakes in the paper and the omissions in the derivations. Additionally, we provide the following clarifications.
>
> 1. On the Definition and Sampling Mechanism of $\tau_t$ (Q1, 2)
>
> We clarify that $\tau_t(s) = \mathbb{E}_{a \sim \pi_t}[\mathbb{I}(\cdot)] $  is theoretically defined as an expectation, which is a  **continuous value**. From an estimation perspective, the subsequent procedural acts as an **unbiased Monte Carlo estimator** of the expectation value. This design aligns with standard RL practices, such as the TD target calculation. Similarly, the variance inherent in our single-sample approach is effectively mitigated by the averaging effect across the mini-batch during SGD training. Crucially, this propose-and-verify sampling procedure is mathematically equivalent to drawing from the theoretical mixed policy. By applying the law of total probability to our accept or reject mechanism, we derive that the actual sampling distribution is:
>
> $$
> \begin{aligned}
> P_{sampling}(a|s) &= P(a|s,Accept) P(Accept|s) + P(a|s,Reject) P(Reject|s)  \\\\
> &=\hat\pi_t(a|s) \tau_t(s) + \pi_{mix,t-1}(a|s) (1-\tau_t (s))
> \end{aligned}
> $$
> , which exactly recovers the definition of $\pi_{mix,t}(a|s)$, thereby rigorously bridging our implementation with the theoretical formulation.
>
> 2. On Design Choices of Policy Evaluation Construction (W2)
>
> The use of $\pi_t$ is justified by the role of the regularization term as a strong imitation term. As long as the reference policy stay within supported region, this regularization term ensures $\pi_t$ remains anchored to supported region. Furthermore, we observe that $\pi_{mix,t}$ struggles to eliminate suboptimal behaviors due to fact that it accounts for a certain weight in the static offline dataset in the mixed buffer, which biases density estimation toward the suboptimal $\pi_\beta$. In contrast, $\pi_t$ is explicitly driven by Q-maximization, empowering it to selectively prioritize high-value actions and escape suboptimal actions more effectively than the $\pi_{mix,t}$.
>
> 3. On Design Choices of Auxiliary Dataset Construction (Q4)
>
> We adopted the progressive append-only strategy primarily to maintain computational feasibility in large-scale settings. Periodically rebuilding $\mathcal{D}'$ would require generating and verifying massive batches of fresh data; since each sample demands a forward pass through both the policy network and the density estimator, this would incur big computational overhead. Our approach efficiently reuses the specific actions $\pi_t$ already generates for policy optimization, minimizing redundant computation. Furthermore, we distinguish our approach from prioritized sampling, which is fundamentally different: prioritized sampling alters the joint **state-action distribution** $p(s,a)$ (biasing state visitation), whereas our method aims to refine the **conditional distribution** $p(a|s)$ (changing actions at the same states). Nevertheless, to fully validate this design, we are currently conducting additional ablations on increasing the batch size of auxiliary data updates.
>
> 4. On Experimental Fairness and Related Work (W1, 3)
>
> We clarify that our tuning strategy is structurally grounded in the fact that CAR generalizes the baseline (SPOT). Specifically, **CAR reduces to SPOT when** $\epsilon_{ood}=0$. Consequently, the first stage of our tuning (optimizing $\lambda$) is strictly equivalent to optimizing the baseline itself to its peak performance. By fixing this optimal baseline configuration and then tuning only $\epsilon_{ood}$, we rigorously isolate the marginal performance gain attributable solely to our proposed coverage-aware mechanism, rather than grid search like CPI or iTRPO. This ensures a fair comparison by demonstrating that our gains persist even when constrained to the baseline's optimal regularization schedule. Furthermore, we will include a discussion of In-sample learning (IQL, SQL) in our paper.

---

### Official Review · Reviewer_o3V3 · 2025-10-31

**Soundness:** 2
**Presentation:** 3
**Contribution:** 2
**Rating:** 2
**Confidence:** 3

**Summary:**

The paper introduces an offline RL framework CAR. The key idea is to design a reference policy that incorporates information from the learned policy (which reflects the current improving signal) and the behavior policy (which provides support information). Benchmarks show potential improvement of proposed framework CAR when compared with other alternatives.

**Strengths:**

The paper is clearly written and the idea makes sense and intuitive.
The benchmarks show potential improvement of proposed framework CAR when compared with
other alternatives. Some theoretical analysis are included.

**Weaknesses:**

The topic is interesting, however the contribution is limited due to the following reasons:
1) it is not clear whether problem formulation introduces any theoretical challenges for analysis.
2) The improvements are not very significant for the benchmarks presented if variances are considered
give the small number of samples (only 5).

**Questions:**

Whether the problem formulation introduces any theoretical challenges for analysis (such as establish Theorem 1 &2)?
Whether the two key hyperparameters change over the training process and task dependent?

---

> ### Author Response · Authors · 2025-11-25
>
> Thank you for your concerns and the effort you have put into our work. We provide the following clarifications.
>
>
> 1. On the Theoretical Challenges (W1 \& Q1)
>
> We clarify that the primary theoretical challenge in **large state space offline RL** lies in the gap between strict theoretical assumptions (e.g., full coverage) and practical algorithms. Most prior works simply **assume** coverage holds, which trivializes the analysis but invalidates the bounds in practice. The challenge our formulation addresses is **how to construct a policy that structurally guarantees** satisfaction of these pessimistic assumptions. Our Theorem 1 is not a off-the-shelf coverage condition. It verifies that our formulation (designed reference policy) successfully overcomes this challenge, **turning the standard coverage assumption into a guaranteed property**. To address the concern on analytical depth, in our response 2 to Reviewer zkx1, we provide a novel theorem that derives the performance bound as an explicit function of $\epsilon$ and analytically solves for the optimal threshold $\epsilon$.
>
> 2. On Experimental Settings (W2 \& Q2)
>
> We confirm that the two key hyperparameters ($\lambda$ and $\epsilon_{ood}$) are task-specific and remain **fixed** throughout the offline training process. Furthermore, in order to further ensure the statistical validity of our results, we are currently running additional experiments to increase the seed count from 5 to 10.

---

> > ### Comment · Reviewer_o3V3 · 2025-11-26
> >
> > Thanks for the authors for the clarification. I would like to keep more scores.

---

### Official Review · Reviewer_zkx1 · 2025-10-31

**Soundness:** 3
**Presentation:** 2
**Contribution:** 2
**Rating:** 2
**Confidence:** 4

**Summary:**

This paper proposes Coverage-Aware Reference Policy (CAR) for offline reinforcement learning (RL).
The key idea is to decouple policy improvement and conservatism via a “propose-and-verify” mechanism:
the learned policy proposes actions, and a verifier checks data support to form a progressively improving reference policy.
Theoretically, the authors provide a coverage guarantee and a performance lower bound; empirically, they report SOTA results on D4RL benchmarks and good online fine-tuning performance.

**Strengths:**

Clear motivation: The paper identifies a real issue in offline RL—entangling improvement with conservatism—and provides a plausible solution.

Theoretical analysis: Two theorems (coverage guarantee and performance bound) are clearly stated and mathematically sound at a conceptual level.

Empirical results: The experiments are extensive, covering MuJoCo, AntMaze, and Adroit, with consistent performance improvements over strong baselines (e.g., CQL, SPOT, CPI).

Readable structure: The paper is well organized and easy to follow, with clear algorithmic pseudocode and hyperparameter analysis.

**Weaknesses:**

Incremental novelty:
The “propose-and-verify” mechanism is essentially a variant of support-constrained sampling as used in prior works such as OSC (Gao et al., 2025) and SPOT (Wu et al., 2022).
The core modification—masking out-of-distribution actions and reweighting with a dynamic threshold—feels like a minor algorithmic variation rather than a conceptually new principle.

Lack of theoretical depth beyond existing work:
Theorems 1 and 2 closely mirror prior proofs on coverage guarantees (e.g., BEAR, CPI, SPOT). The results do not introduce fundamentally new analytical tools or insights.

Empirical claims are overstated:
Although the table reports high scores, performance margins are small or within error bars on most MuJoCo tasks.
There is no statistical significance analysis, and no ablation isolating the specific effect of “decoupling” vs. the density model choice.

Ambiguity in verification mechanism:
The implementation of the “verify” step is heuristic—based on a VAE reconstruction threshold rather than a formal likelihood test.
This weakens the theoretical connection between the coverage guarantee and the practical algorithm.

Over-claiming of generality:
The method is presented as a “decoupling framework” for general offline RL, but its applicability beyond VAE-based density regularization (e.g., to actor-critic with other generative models) is unclear.

Writing and clarity issues:
Several grammatical errors and unclear sentences (e.g., “conservatism is enforced by a verification step that guarantees these actions are within the data’s support”) reduce professionalism.
Some equations (e.g., Eq. 7, Eq. 9) lack intuition or clear derivation paths in the main text.

**Questions:**

How does CAR differ principally from OSC (Gao et al., 2025) or CPED (Zhang et al., 2023)?
Please provide a conceptual comparison table.

Can the authors show cases where the decoupling significantly improves learning stability, not just final score?

Is the threshold ϵ fixed or adaptive? How sensitive is the algorithm to this choice beyond Figure 2(b)?

Please clarify whether the verifier can reject all actions in a batch, and what happens in that case.

It would help to include computational complexity comparisons in terms of FLOPs or wall time, not just ms/step.

---

> ### Author Response · Authors · 2025-11-25
>
> We sincerely thank you for your concerns. We will clarify the main points regarding our concept, theory, and nuanced implementation details of our method.
>
> 1. On the Novelty and Distinction from Prior Works (W1 \& Q1)
>
> We position CAR within the shared **support constraint** framework for **large state space** offline RL, but fundamentally distinguish its core principle. Prior works (e.g., SPOT/CPED) convert constraints into **regularization** terms, which inadvertently forces the policy to **strictly mimic $\pi_\beta$ even within supported regions**, leading to over-conservatism; meanwhile, OSC applying a hard clip (indicator function) to the regularization, which hinders optimization with discontinuous gradients. CAR fundamentally redefines the objective: instead of tracking $\pi_\beta$, we aim to mimic an improved reference policy that remains supported. The propose-and-verify mechanism is the concrete implementation of this dynamic reference, enabling the policy to deviate from $\pi_\beta$ while rigorously respecting support constraints.
>
>
> 2. On Theoretical Consistency and Verification (W2, 4)
>
> We introduce a new performance bound **competing the optimal policy**: $J(\pi^\star) - J(\pi_{mix,t}) \le \mathcal{O}(C^\star\epsilon) + \mathcal{O}\left( \frac{V_{\max}}{1-\gamma}\sqrt{\frac{\ln(|\mathcal{F}|/\delta)}{n\epsilon}} \right)$, where $C^\star$ is the single-policy concentrability coefficient. The first term captures the cost of rejecting optimal actions in low-density regions, while the second captures function approximation error. Minimizing this trade-off yields the theoretically **optimal** **$\epsilon$** $\propto \left( \frac{V_{\max}\sqrt{\ln(|\mathcal{F}|/\delta)}}{C^*(1-\gamma)\sqrt{n}} \right)^{2/3}$, proving that $\epsilon$ is not an arbitrary heuristic but a principled hyperparameter necessitating task-specific adaptation. In terms of implementation, the VAE provides a tractable approximation of the density via the Evidence Lower Bound (ELBO). Thresholding the ELBO is thus a mathematically grounded proxy for the theoretical support constraint.
>
> 3. On Empirical Significance and Ablation (W3, 5 \& Q2, 4)
>
> We demonstrate that performance gains are strictly attributable to our 'decoupling' framework rather than the density model choice. First, **SPOT** serves as a direct coupled baseline (effectively a **special case of CAR with $\epsilon_{ood}=0$**) using the identical VAE backbone; the substantial performance gap across all tasks isolates and validates the specific contribution of our propose-and-verify mechanism. Second, we validate Density Model Independence (Fig. 2d, Fig. 7) by showing CAR consistently functions across Gaussian, VAE, and Diffusion backbones, confirming the principle is generalizable. Finally, addressing the rejected batch concern, our fallback strategy ($\pi_{mix,t+1} = \pi_{mix,t}$) ensures stability: even if all proposed actions are rejected, the reference policy simply remains **unchanged**, implying no erroneous update occurs at the current step.
>
> 4. On Sensitivity and Computational Complexity (Q3, 5)
>
> We confirm $\epsilon$ is a **fixed** hyperparameter for each task. To demonstrate sensitivity, Figure 2(a) (rows) explicitly illustrates the performance sensitivity to varying $\epsilon$. Regarding computational complexity, we will include a FLOPs comparison in the revision to provide hardware-independent verification.

---

### Official Review · Reviewer_r5oJ · 2025-11-01

**Soundness:** 3
**Presentation:** 3
**Contribution:** 3
**Rating:** 6
**Confidence:** 3

**Summary:**

This work proposes CAR, a coverage-aware reference policy in which the learned policy proposes actions and a verifier confirms whether the data are in-distribution before augmenting the reference policy. The core idea builds on designing a sampling mechanism to avoid directly estimating the normalization term. Additionally, a simple theoretical analysis supports the effectiveness of CAR. In the experiment, CAR consistently outperforms other baseline methods.

**Strengths:**

1. The paper is clearly structured, allowing readers to follow the overall flow and reasoning easily.

2. CAR identifies and addresses the limitations of previous approaches that relied on reference policies. Moreover, employing various behavior policy estimation methods demonstrates the robustness and validity of its claims.

3. CAR outperforms 7 baseline methods on the MuJoCo, AntMaze, and Adroit tasks.

**Weaknesses:**

1. For Table 1, only RORL is trained for 3M gradient steps, which might obscure the superior performance of CAR on the MuJoCo tasks. It would be helpful to reproduce the normalized score of RORL with 1M gradient steps for a fair comparison.

2. Compared to the offline setting, where CAR is evaluated against 7 baseline methods, the offline-to-online experiments include comparisons with only 2 algorithms (SPOT and Cal-QL). A more detailed explanation and broader comparison would strengthen the analysis. For example, RLPD [1] and PEX [2].

[1] Ball, Philip J., et al. "Efficient online reinforcement learning with offline data." International Conference on Machine Learning. PMLR, 2023.

[2] Zhang, Haichao, We Xu, and Haonan Yu. "Policy expansion for bridging offline-to-online reinforcement learning." arXiv preprint arXiv:2302.00935 (2023).

**Questions:**

1. When fixing the VAE model during online fine-tuning, are you using only the online dataset, or a mixture of both offline and online data? If only the online data are used, the fixed VAE model may experience a distributional shift, leading to potential forgetting before it adapts again. On the other hand, if mixed data are used, the offline data may dominate the updates, making it difficult for the model to sufficiently reflect the influence of the online data.

2. For Figure 2, the values of $\epsilon_{ood}$ and $\lambda$ appear to be highly sensitive. In Table 2, the balance coefficient ranges from 0.05 to 38, which seems excessively wide. Moreover, the key hyperparameter values of CAR for each dataset are not provided.

---

> ### Author Response · Authors · 2025-11-25
>
> We sincerely thank you for your concerns. We will clarify the main points regarding our concept, theory, and nuanced implementation details of our method.
>
>
> 1. On Fair Comparison and Baselines (W1, 2)
>
> We agree with the importance of a strictly fair comparison and we will report the performance of RORL with 1M gradient steps in the paper. Additionally, to address the concern on baseline coverage, we have incorporated RLPD and PEX into our offline-to-online experiments.
>
>
> 2. On Data Usage during Online Fine-tuning (Q1)
>
> We confirm that we retain the **full offline dataset** during fine-tuning, consistent with the SPOT baseline. Addressing the concern about VAE distribution shift, we conducted additional experiments comparing: CAR-(1) Updating VAE on mixed data ($\mathcal{D} \cup \mathcal{D}' \cup \mathcal{D}_{online}$) with **constant** $\lambda$, and CAR-(2) Updating VAE with **decaying** $\lambda$.
> Results are shown in the following table, CAR-(2) achieves the best performance, surpassing our original fixed-VAE setting (CAR), while CAR-(1) performs worst. **Insight**: Decaying $\lambda$ is critical; it gradually relaxes the constraint, mitigating the negative impact of potential VAE bias (dominated by suboptimal offline data), whereas constant $\lambda$ over-constrains the agent. We will report these improved results in paper.
>
> | AntMaze |  | CAR | CAR-(2) | CAR-(1) |
> | :--- | :--- | :---: | :---: | :---: |
> | antmaze-umaze                | 99.2 $\rightarrow$ | 99.6±0.4 | 99.8±0.5 | 99.4±0.5 |
> | antmaze-umaze-diverse   | 95.6 $\rightarrow$ | 99.4±0.7 | 99.7±0.8 | 98.9±1.2 |
> | antmaze-medium-play      | 93.6 $\rightarrow$ | 99.0±0.8 | 99.2±1.0 | 98.7±1.1 |
> | antmaze-medium-diverse | 89.8 $\rightarrow$ | 99.1±0.7 | 99.1±1.1 | 97.8±1.6 |
> | antmaze-large-play           | 78.0 $\rightarrow$ | 96.9±1.1 | 97.5±1.2 | 95.1±2.2 |
> | antmaze-large-diverse      | 77.8 $\rightarrow$ | 96.5±1.2 | 97.1±0.9 | 94.5±2.8 |
> | Average                             | 89.0 $\rightarrow$ |  98.4   |  98.7   |   97.4    |
>
>
> 3. On Hyperparameter Sensitivity (Q2)
>
> We clarify that task-specific parameters are currently provided in our code and will be tabulated in the paper. Regarding parameter sensitivity, our new theoretical analysis (see Response 2 to Reviewer zkx1) proves that the optimal threshold satisfies $\epsilon\propto (n^{-1/2} / C^\star)^{2/3}$. Since the dataset size $n$ and optimal policy concentrability $C^\star$ vary significantly across tasks, the wide range of hyperparameters is theoretically justified and necessary for adaptation.

---

### Meta-Review · Area_Chair_FoPy · 2026-01-06

**Summary:**

The reviewers expressed mixed opinions, with one weak accept (Rating 6) and three rejects (Rating 2). The concerns driving the negative recommendations cluster around three major themes:

1. Novelty and conceptual contribution: Multiple reviewers (zkx1, o3V3) viewed the "propose-and-verify" mechanism as a minor algorithmic variation of existing support-constrained methods (SPOT, OSC, CPED) rather than a fundamentally new principle. The core decoupling idea was seen as insufficiently distinct from prior regularization-based approaches.
2. Theoretical rigor and clarity: Theorems were criticized as mirroring prior work (BEAR, CPI) without introducing fundamentally new analytical tools (zkx1, o3V3). A disconnect was identified between the theoretical formulation (mixed policy with coefficient) and the actual implementation (x78z). The proof of Theorem 1 was flagged for typos, logical leaps, and implicit assumptions that need clarification (x78z). The use of a heuristic VAE reconstruction threshold (rather than a formal likelihood test) was seen as weakening the theoretical connection (zkx1).
3. Empirical evaluation and fairness: Performance margins were often within error bars, with no significance testing (zkx1). The small sample size (5 seeds) was deemed insufficient (o3V3).

While reviewers acknowledged the paper's clear writing and strong baseline results, the primary concerns centered on insufficient conceptual novelty and questionable theoretical depth that undermined confidence in the method's claimed contributions.

**Reviewer Concerns:**

The rebuttal successfully addresses most implementation clarifications, fairness issues, and theoretical framing concerns. For example:
* Theoretical depth & novelty: Authors clarified CAR's core principle differs from SPOT/CPED (regularization-based) and OSC (hard clip) by dynamically improving the reference policy rather than mimicking $\pi_{0}$. The authors also explained that the theoretical contribution is converting coverage assumptions into guaranteed properties, addressing the gap between theory and practice.
* The authors demonstrated density model independence across Gaussian, VAE, and Diffusion backbones (Fig. 2d, Fig. 7), validating framework generality.

However, it falls short on providing concrete new results (RORL step mismatch (r5oJ W1), baselines for offline-to-online (r5oJ W2), hyperparameter transparency (r5oJ Q2), 10 seeds results) and detailed corrections to theoretical derivations. I think it's unlikely that all the reviewers would increase score from 2 to 6.

**Reviewer Scores:**

Reviewer zkx1: Initial Score: 2 -> Likely New Score: 4

While the rebuttal makes a strong effort to reposition CAR's novelty (improving reference policy vs. mimicking $\pi_0$) and adds theoretical depth (optimal $\epsilon$ derivation), this reviewer entered with high confidence and deep skepticism about fundamental contribution. The rebuttal's claims remain promises rather than verified revisions—no corrected proofs, no FLOPs data, and no concrete statistical significance analysis are provided. The reviewer would likely acknowledge the effort but remain unconvinced that the work exceeds the incremental-variation bar for a top-tier venue.

Reviewer o3V3: Initial Score: 2  -> Likely New Score: 2 (no change)

The reviewer's follow-up comment explicitly states "I would like to keep more scores," indicating a wait-and-see stance. Since the  reviewer's primary empirical concern (sample size/variance) is only promised to be addressed (10 seeds "currently running"), not demonstrated, they would likely hold their position until seeing actual results. The theoretical clarification is appreciated but insufficient to overcome the statistical robustness concern.

Reviewer x78z: Initial Score: 2 -> Likely New Score: 4

The rebuttal excellently resolves several specific technical questions (σ sampling mechanism, tuning fairness) and promises to add missing related work. However, this confident reviewer raised precise issues about proof clarity/typos and auxiliary dataset ablations. The rebuttal only acknowledges these problems without showing corrected proofs or ablation results. High-confidence reviewers typically demand to see the actual fixes before raising scores, so I think the reviewer would likely soften slightly but remain below the acceptance threshold.

---

### Decision · Program_Chairs · 2026-01-26

Reject